# Stream RAG: Instant and Accurate Spoken Dialogue Systems with Streaming Tool Usage

**Siddhant Arora** [1 2]  **Haidar Khan** [1]  **Kai Sun** [1]  **Xin Luna Dong** [1]  **Sajal Choudhary** [1]  **Seungwhan Moon** [1]
**Xinyuan Zhang** [1]  **Adithya Sagar** [1]  **Surya Teja Appini** [1]  **Kaushik Patnaik** [1]  **Sanat Sharma** [1]  **Shinji Watanabe** [2]
**Anuj Kumar** [1]  **Ahmed Aly** [1]  **Yue Liu** [1]  **Florian Metze** [1]  **Zhaojiang Lin** [1]

## Abstract

End-to-end speech-in, speech-out dialogue systems are emerging as a powerful alternative to traditional ASR–LLM–TTS pipelines but remain prone to hallucinations due to limited factual grounding. While text-based dialogue models have effectively mitigated this issue through tools such as web search APIs, extending such capabilities to speech-in, speech-out systems remains underexplored. A key challenge is that tool integration increases latency, disrupting conversational flow. To mitigate this, we propose Streaming Retrieval-Augmented Generation (*Stream RAG*), a novel framework that reduces latency by predicting tool queries in parallel with user speech, even before the user finishes speaking. Specifically, we develop a post-training pipeline that teaches the model when to *issue tool calls* and how to generate spoken summaries using retrieved text results, thereby improving both *accuracy and responsiveness*. To evaluate our approach, we construct AudioCRAG, a benchmark created by converting queries from the publicly available CRAG dataset into speech form. Experimental results show that *Stream RAG* improves QA accuracy by over 20.0% absolute on AudioCRAG and achieves state-of-the-art performance, including outperforming cascaded systems, on the SLUE-SQA benchmark, while reducing latency by up to 57%. *Stream RAG* is modality-agnostic and can be applied equally to typed input, paving the way for more agentic, real-time AI assistants.

Work done while Siddhant Arora was interning at Meta [1]Meta AI, USA [2]Carnegie Mellon University, USA. Correspondence to: Siddhant Arora <siddhana@meta.com>, Zhaojiang Lin <zhaojiang@meta.com>.

## 1. Introduction

Spoken Dialogue Systems (SDS) are foundational to many everyday technologies, powering intelligent assistants such as Alexa and Siri, interactive voice response systems in customer service as well as mobile phones and wearable devices. Traditionally, SDS have relied on cascaded pipelines composed of multiple modules, each introducing potential points of failure and latency (Glass, 1999; Huang et al., 2024). Recently, end-to-end (E2E) SDS (Xie & Wu, 2024; Nguyen et al., 2023; Meng et al., 2024; Zhang et al., 2024; Arora et al., 2025b) have been proposed, which directly generate spoken responses from speech input within a unified architecture. This E2E approach not only mitigates error propagation across modules but also captures non-phonemic information more effectively, resulting in lower inference time and computational overhead.

Despite these advances, current E2E SDS are fundamentally constrained by their reliance on internalized knowledge from static training data, which often results in responses that lack factual grounding or fail to reflect up to date information. This shortcoming is particularly critical for action-oriented or knowledge-seeking tasks, such as booking hotels or answering questions about current events. In contrast, text-based assistants have begun to overcome these limitations by integrating external tools through Retrieval-Augmented Generation (RAG) (Yang et al., 2024d; Chen et al., 2023a;c; Gao et al., 2024a;b), dynamically retrieving information from sources like web search and knowledge graphs (KG) APIs. Yet, only a few works (Feng et al., 2025b; Chen et al., 2025) have begun to explore the integration of tool use into E2E SDS. A key challenge is that while external tools can improve factual accuracy, invoking them often introduces additional latency, leading to awkward silences that disrupt the natural conversational flow. *How can we trade-off between accuracy and responsiveness for developing SDS that feel both intelligent and natural?*

In this paper, we present, to the best of our knowledge, the *first latency-aware, learned scheduling policy for streaming tool-query generation in E2E speech-in, speech-out dialogue systems*. The key idea is a **Stream RAG** strategy

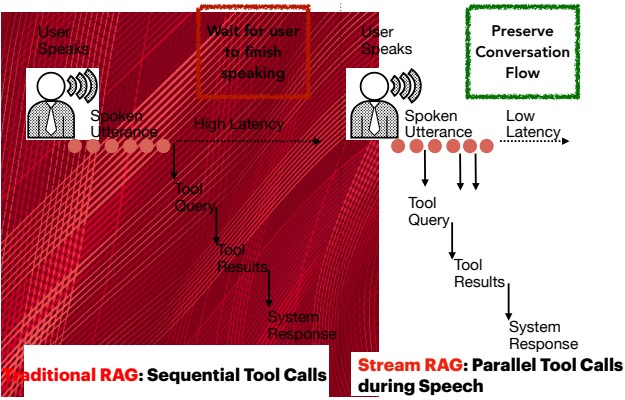

*Figure 1.* Comparison of Traditional RAG with proposed Stream RAG which fires tool queries in parallel with user speech.

that generates tool queries in parallel with user speech, often before the user has finished speaking (Fig. 1). A naive implementation of streaming queries, however, faces two challenges: (1) queries issued from partial speech may be suboptimal, yielding distracting tool outputs and inaccurate responses; and (2) unnecessary tool calls may be triggered, wasting computational resources. We introduce effective modeling techniques to address these challenges and make the following contributions.

**Contribution 1**: We present an empirical study of latency bottlenecks in tool-augmented SDS. Evaluating three state-of-the-art models, Qwen-OMNI (Xu et al., 2025), OpusLM (Tian et al., 2025), and Kimi-Audio (Ding et al., 2025), we show that external tools substantially improve accuracy (e.g., 11.1% → 26.3% for Qwen-OMNI) but increase first-token latency by up to 2.3× (from 1.34s to 5.9s in T. 1). These results highlight a key tension: retrieve-after-endpoint approaches, while effective for accuracy, often struggle to meet the real-time responsiveness requirements.

**Contribution 2**: To address this, we propose *Streaming Retrieval-Augmented Generation* (*Stream RAG*), the *first* framework that empowers the system to trigger tool queries in parallel with user speech. Within this framework, we introduce two novel approaches: (1) *Fixed-Interval Stream RAG*, which issues tool queries at regular intervals during speech input and carefully examines quality of retrieval results on the full query to guarantee response quality, while remaining compatible with any speech-in, speech-out model without post-training; (2) *Model-Triggered Stream RAG*, which post-trains the model to intelligently determine optimal query timing based on the evolving user utterance to save computation resources. Our proposed *Model-Triggered Stream RAG* improves QA accuracy from 11.1% to 34.2% absolute (T. 1) compared to the no-tool baseline, while also reducing first-token latency by 57% (§5.4). On the SLUE-SQA benchmark (Shon et al., 2023), Stream RAG achieves an exact match accuracy of 55.8%, outperforming prior RAG approaches (Chen et al., 2025) and cascaded pipeline,

while responding even before the user finishes speaking, demonstrating gains in both accuracy and latency.

**Contribution 3**: Finally, we introduce AudioCRAG, a benchmark created by recording spoken queries from the CRAG (Yang et al., 2024d) dataset, enabling robust evaluation of tool usage capabilities in speech-in, speech-out systems. We will open source our training code, supporting future research in tool-integrated voice assistants.

## 2. Related studies

### 2.1. Benchmarks for Tool Usage

Recent advances in benchmarking text-based dialogue systems for tool usage (Chen et al., 2023b; Ouyang et al., 2025; Cheng & Dou, 2025) have primarily focused on evaluating factual question answering and task completion (detailed discussion in Appendix A.8, A.9). CRAG (Yang et al., 2024d) is a leading example, featuring 4,409 question-answer pairs and providing mock APIs for both web and KG search. Recent benchmarks (Meta CRAG-MM Challenge Organizers, 2025; Ma et al., 2024; Jang et al., 2025) have extended tool-augmented dialogue evaluation to multimodal input. While these benchmarks have advanced the evaluation of tool-augmented dialogue systems, they remain largely limited to text-based outputs and do not fully address the unique challenges presented by speech-in, speech-out systems.

### 2.2. E2E Spoken Dialogue Systems

Several E2E SDS (Xu et al., 2025; Xie & Wu, 2024; Arora et al., 2025a) have recently been introduced, demonstrating impressive semantic understanding and high audio quality in their responses. Concurrent submissions by overlapping authors explore foundation models (Chen et al., 2026; Tian et al., 2026) for open-ended audio understanding and generation (See S. A.17 for details). However, these works rely exclusively on internalized knowledge and have not been trained or evaluated for their ability to *use external tools*, a capability that is critical for factual grounding and action-oriented dialogue. Early efforts to incorporate external tools via RAG have primarily focused on spoken language understanding (Wang et al., 2024; Yang et al., 2024b;a) and audio captioning (Ghosh et al., 2024). A related line of work (Feng et al., 2025a; Min et al., 2025; Sun et al., 2025) investigates E2E RAG for direct speech-to-text retrieval, leveraging multimodal embeddings to retrieve relevant textual information from spoken queries. More recently, web-based systems have integrated tool usage into speech-in, speech-out settings using cascaded pipelines (Maben et al., 2025). Other works (Feng et al., 2025b) like WavRAG (Chen et al., 2025), extends RAG to E2E speech-in, speech-out models using audio–text fusion. However, a key limitation shared by prior approaches is that they perform retrieval only after

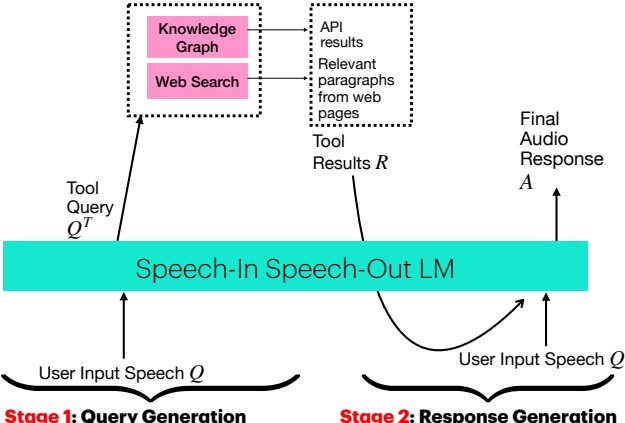

*Figure 2.* Formulation for integrating tool usage in E2E speech-in, speech-out dialogue systems using a two-stage inference approach.

endpoint detection. This retrieve-after-endpoint paradigm introduces several seconds of additional latency when tool calls require queries over 100K web documents, resulting in conversational delays that exceed established human turn-taking thresholds (S. A.2). Hence, we introduce Stream RAG, the *first* framework that enables models to issue tool queries during ongoing speech, effectively masking external API latency while preserving response quality.

## 3. Methodology

A RAG spoken conversation system takes an audio question $Q$ as input and outputs a spoken answer $A$. Answers are generated by speech-in, speech-out models, leveraging the model's internal knowledge and information retrieved from external sources as shown in Figure 2. To incorporate external information, the model needs to formulate a tool query $Q^T$ to retrieve relevant results $R$ from an external tool $T$.

### 3.1. Tool Integration for Speech-in, Speech-Out LLMs

Figure 2 illustrates our proposed formulation for integrating external tools into speech-in, speech-out systems. We use a two-stage inference approach: *Query Generation* and *Response Generation*. In the Query Generation stage, the system processes an audio question and generates queries for each external tool to retrieve relevant information by maximizing the posterior distribution $P(Q^T|Q)$ (Examples of generated queries are provided in T. 15 in the Appendix.). Each tool call invokes a standard retrieval pipeline consisting of (i) top-$k$ document retrieval, (ii) document chunking, and (iii) reranking over retrieved chunks. Notably, the chunking and reranking steps **dominate first-token latency** (T. 4), creating multi-second delays that disrupt conversational flow. In the Response Generation stage, the retrieved results $R$ from these tools are combined with the audio question and input into the model to generate the final spoken response by maximizing the posterior distribution $P(A|Q, R)$.

By conditioning the final output generation on the input audio, this formulation not only provides a simple and effective mechanism for interacting with text-based APIs, but also preserves the key advantages of speech-in, speech-out systems, mitigating error propagation and enabling the model to capture non-phonemic information more effectively.

### 3.2. Stream RAG

RAG-based systems, as proposed in S. 3.1, can significantly improve factual accuracy by incorporating external tools. However, these tool calls often introduce substantial latency, which is particularly problematic in speech-in, speech-out applications where users expect rapid, conversational responses and even brief silences can disrupt the natural flow of dialogue. One way to address this challenge lies in the nature of audio inputs, which arrive as a continuous stream. This streaming property enables tool calls to be initiated before the user has finished speaking, offering a unique opportunity to mitigate latency.

To minimize user-perceived latency, we introduce *Stream RAG*: the first framework to generate and issue tool queries in parallel with ongoing audio input. This novel approach is built on three key design components: ① Trigger: When to initiate a new tool query; ② Threads: The number of parallel tool query threads; ③ Reflector: The module that determines whether intermediate tool results are sufficient for generating the final output. By exploring different design choices for each component, we introduce two complementary approaches for streaming tool query generation in the following subsections: a fixed-interval trigger method and a model-based trigger method.

**Challenges of Streaming Tool Invocation.** Unlike conventional RAG systems, which perform retrieval only after the complete user utterance is available, Stream RAG operates under partial observability: retrieval decisions must be made while the user is still speaking. This introduces several challenges that are absent in endpoint-only retrieval. First, the system must determine when partial speech contains sufficient information to justify a tool call versus when it should wait for additional context. Second, premature or incorrect tool queries may lead to error propagation, where early retrieval mistakes negatively affect subsequent decisions. Third, streaming retrieval introduces an explicit latency-accuracy tradeoff: issuing tool calls earlier can reduce response delay but may harm retrieval quality if insufficient information is available. While recent streaming speech dialogue systems such as Moshi (Défossez et al., 2024), NTPP (Wang et al., 2025), and LLaMA-Omni (Fang et al., 2024) focus on low-latency response generation, they do not address the problem of generating retrieval queries from partial speech or conditioning responses on retrieved external evidence. Stream RAG therefore studies a comple-

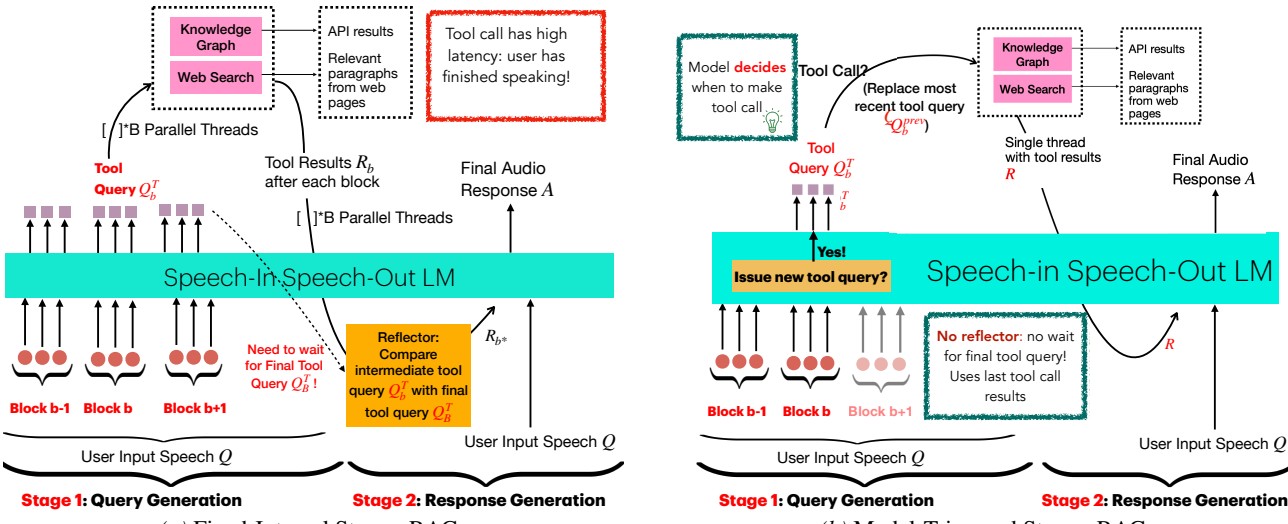

*(a)* Fixed-Interval Stream RAG.    *(b)* Model-Triggered Stream RAG.

*Figure 3.* Proposed formulations for streaming tool query generation, referred to as *Stream RAG*, to minimize user-perceived latency in speech-in, speech-out systems. (a) Fixed-Interval Stream RAG: tool calls are triggered at fixed intervals and evaluated by a reflector module that must process the final speech block, creating a latency bottleneck. (b) Model-Triggered Stream RAG: the model autonomously decides when to make tool calls, eliminating the reflector and directly utilizing the most recent tool results for response generation.

mentary capability: streaming tool invocation under partial observations.

### 3.2.1. FIXED-INTERVAL STREAM RAG

In this approach, the *trigger* is set to fire tool calls at fixed chunk intervals during audio input. The input speech $Q$ is divided into a sequence of $B$ blocks, $Q = \{Q_b \mid b = 1, \ldots, B\}$, with each block containing $N_{\text{block}}$ frames. To approximate $P(Q^T|Q)$ as described in S. 3.1, we follow a block-wise prediction strategy. In this approach, after processing each audio block $b$, the model predicts a tool query $\hat{Q}_b^T$ by conditioning on the input speech accumulated up to block b, specifically $Q_{1:b}$:

$$\hat{Q}_b^T = \arg\max_{Q_b^T} P(Q_b^T|Q_{1:b}) \qquad (1)$$

This strategy results in $B$ parallel tool call threads running simultaneously (see Figure 3a), where each thread generates a tool query prediction $\hat{Q}_b^T$ for its corresponding block $b \in [1, B]$. The tool queries $\hat{Q}_b^T$ generated after each block are then stored in cache. Given the high latency of tool calls (See T. 7), users typically complete their utterances before tool responses are ready. Upon utterance completion, an explicit *reflector* module "reflect()" evaluates the cached intermediate queries $\hat{Q}_b^T$ against final query $\hat{Q}_B^T$ to determine whether an early intermediate tool call provides sufficient information to answer the user's question $Q$. The reflector module systematically evaluates all intermediate queries in the cache and identifies the *earliest* sufficient tool call $b^\star$ where the intermediate query $\hat{Q}_b^T$ will give the same result

as final query $\hat{Q}_B^T$ as shown:

$$b^\star = \min\{b \in [1, B] \text{ where reflect}(\hat{Q}_b^T, \hat{Q}_B^T) = \text{True}\}. \qquad (2)$$

All subsequent parallel tool calls after $b^\star$ are promptly terminated, and the retrieved results $R_{b^\star}$ from this intermediate call $\hat{Q}_{b^\star}^T$ are used to generate the final spoken response $A$ by maximizing the posterior distribution $P(A|Q, R_{b^\star})$ (instead of $P(A|Q, R)$ in S. 3.1). We employ a reflector module that uses simple yet effective heuristics: (a) if the top 5 web documents for an intermediate web query match those of the final web query, and (b) if the KG results for intermediate and final KG queries are identical. These heuristics ensure that the information retrieved from an early tool call using $\hat{Q}_{b^\star}^T$ is consistent with what would have been obtained by waiting for the final tool call using $\hat{Q}_B^T$, thereby providing a strong quality guarantee. Since most tool call latency arises from tool results generation, specifically the chunking and reranking steps (S. 3.1), these checks enable modest but consistent latency savings without compromising model performance. A key advantage of this strategy is its plug-and-play nature: it requires no additional post-training and can be directly applied at inference time across a variety of architectures. However, this simplicity comes with trade-offs. First, generating parallel tool calls at every fixed interval (Eq. 1) increases computational overhead, potentially limiting deployment on resource-constrained devices like wearables. Second, reliance on an external reflector module (Eq. 2), which must process the final input block to confirm the last query and judge whether intermediate results are sufficient, limits potential latency savings. This dependency means the approach **does not fully eliminate endpoint-dependent latency**, response generation still waits for utterance completion. Overall, Fixed-Interval Stream RAG is therefore

intentionally positioned as a *simple, training-free* baseline to isolate the effects of parallel tool execution rather than to maximize latency reduction.

### 3.2.2. MODEL-TRIGGERED STREAM RAG

To address the limitations of Fixed-Interval Stream RAG and further optimize both efficiency and responsiveness, we propose *Model-Triggered Stream RAG*. Unlike heuristic scheduling, this approach learns a policy over tool invocation actions under partial observability, where decisions are made based on incomplete, evolving speech input. The trigger is learned: the model is trained to autonomously determine the optimal moments to initiate tool queries, issuing a query only when it encounters new or additional information as shown in Figure 3b. In this formulation, the model receives user input in fixed chunk intervals as before. The model can either: ① Predict NO_QUERY if a new tool query is unnecessary, or ② Generate a new tool query. To make this decision, the model conditions on both the accumulated input speech $Q_{1:b}$ (see Eq. 1) as well as the most recent tool query $\hat{Q}_b^{\text{prev}} = \hat{Q}_{\max\{\,i<b:\hat{Q}_i^T \neq \text{NO\_QUERY}\,\}}^T$:

$$\hat{Q}_b^T = \arg\max_{Q_b^T} P(Q_b^T|Q_{1:b}, \hat{Q}_b^{\text{prev}}) \qquad (3)$$

When a new query $\hat{Q}_b^T \neq \text{NO\_QUERY}$, the system immediately terminates any ongoing tool calls for the previous query $\hat{Q}_b^{\text{prev}}$, ensuring that only a *single tool call thread* runs at any given time. (Examples of generated $\hat{Q}_b^T$ are provided in T. 16 in the Appendix.) This approach offers several key advantages. First, it effectively eliminates redundant parallel threads and significantly reduces computational overhead. This is especially important for deployment on resource-constrained devices. Second, this formulation removes the need for an external *reflector* (Eq. 2) module. The model confidently relies on the results $R$ from the most recent tool call using $\hat{Q}_B^{\text{prev}}$ to generate the spoken response $A$ by maximizing $P(A|Q, R)$, reducing system complexity.

**Post-training**: To train the model, we transform text-based tool usage benchmarks into spoken format (See S. A.14). Let the automatic speech recognition (ASR) transcript of audio question be $X^{\text{asr}}$. Word-level timestamps are computed using a pre-trained ASR model. For each partial ASR transcript $X_b^{\text{asr}}$ up to block $b$, we generate corresponding queries for each tool $\overline{Q_b^T}$ using an LLM as pseudo ground truth (GT). To create effective training labels that teach the model when to trigger new queries, we employ a similarity-based labeling strategy where we compare the current pseudo GT query $\overline{Q_b^T}$ with the most recent non-empty tool query label $\hat{Q}_b^{\text{prev}}$ (see Eq. 3) before block $b$. Our labeling function assigns the training label $\hat{Q}_b^T$ (Eq. 3) for the tool query after block $b$ as follows: ① when the current query is sufficiently similar to the previous query (as determined by manually defined

heuristics $f(\cdots)$), we assign the special label NO_QUERY to teach the model that no new tool call is needed. ② When the queries are sufficiently different, we assign the actual pseudo ground truth query $\overline{Q_b^T}$ as the label to teach the model to trigger a new tool call:

$$\hat{Q}_b^T = \begin{cases} \text{NO\_QUERY}, & \text{if } f(\overline{Q_b^T}, \hat{Q}_b^{\text{prev}}) = \text{True}, \\ \overline{Q_b^T}, & \text{else .} \end{cases} \qquad (4)$$

For KG queries, we assign a NO_QUERY label when the current query exactly matches the previous one. For web queries, we assign a NO_QUERY label if the top five retrieved documents for the current query remain unchanged from the previous query. We employ a multi-task fine-tuning strategy targeting two key capabilities. First, we train the model on Streaming Tool Query Generation by optimizing $P(Q_b^T|Q_{1:b}, \hat{Q}_b^{\text{prev}})$ for $b \in [1, B]$, enabling intelligent decisions about when to trigger tool queries. Second, we fine-tune on Response Generation by optimizing $P(A|Q, R)$ (S. 3.1) to improve the intelligibility of the speech output.

An important aspect of our post-training is enhancing the model's ability to recover from errors in intermediate query predictions. For example, when presented with the audio question, "Who founded Rare Beauty in 2019?", we observed that an initial misinterpretation of $\hat{Q}_b^{\text{prev}}$, such as "Red Bull founder", can lead the model to subsequently generate NO_QUERY labels, effectively halting further attempts to retrieve the correct information. This issue arises because, during training, the model is always provided with correct previous query $\hat{Q}_b^{\text{prev}}$, whereas during inference, it may make errors due to partial utterances $Q_{1:b}$ being ambiguous. Thus the model lacks the ability to recover from such mistakes during inference. To improve robustness under partial observability, we deliberately inject negative samples during post-training by substituting the previous query $\hat{Q}_b^{\text{prev}}$ in Eq. 4 with incorrect ones $Q_b^{\text{neg}}$. Crucially, when we perform negative sampling, we fall back to the pseudo ground truth query $\overline{Q_b^T}$ as the training label:

$$(\hat{Q}_b^T, \hat{Q}_b^{\text{prev}}) = \begin{cases} (\hat{Q}_b^T, \hat{Q}_b^{\text{prev}}), & \text{with probability } 0.9, \\ (\overline{Q_b^T}, Q_b^{\text{neg}}), & \text{with probability } 0.1. \end{cases} \qquad (5)$$

This novel strategy teaches the model to recover from errors in intermediate query prediction, thereby maintaining accuracy (see T. 14 for ablation) while achieving latency savings. To further validate robustness to misfired early tool calls, we conduct stress tests with intentionally misleading previous queries (see §A.3), where the model achieves 65.4% recovery accuracy, demonstrating its ability to self-correct as more speech becomes available.

*Table 1.* Performance comparison of Qwen-OMNI, OpusLM, and Kimi Audio across Closed Book, Open Book, and Stream RAG settings on both AudioCRAG-Synthetic (Syn.) and AudioCRAG- Human (Hum.). For reference, strong text-in/text-out LLM baselines achieve comparable accuracy on CRAG (Yang et al., 2024d) (e.g., GPT-4 Turbo: 43.6%, LLaMA-3-8B Instruct: 32.1%). Stream RAG is not applied to Kimi-Audio as it can handle only a restricted length of tool result references (S. A.13). *: OpusLM currently does not support taking tool result references in speech-out settings in zero-shot. Note: With vLLM optimization (§5.4), first-token latency further reduces to 1.36s for AudioCRAG-Human. We additionally compare against a post-trained sequential RAG baseline to disentangle the effects of post-training and streaming retrieval in Table 6.

| Setting | Ref length | Model | Accuracy | | Latency | | | |
| | | | | | First Token (s) | | Tool Use (s) | |
| | | | Syn. | Hum. | Syn. | Hum. | Syn. | Hum. |
| --- | --- | --- | --- | --- | --- | --- | --- | --- |
| Closed Book | 0 | Qwen-OMNI | 11.1 | 13.1 | 1.34 | 1.24 | ✗ | ✗ |
| | 0 | OpusLM | 18.4 | 15.5 | 5.67 | 7.07 | ✗ | ✗ |
| | 0 | Kimi Audio | 16.7 | 16.0 | 0.85 | 0.89 | ✗ | ✗ |
| Open Book | 23K | Qwen-OMNI | 26.3 | 26.9 | 5.90 | 5.40 | 3.37 | 3.37 |
| (S. 3.1) | 15K | OpusLM* | 0.0 | 0.0 | 9.05 | 10.44 | 3.97 | 3.97 |
| | 500 | Kimi Audio | 21.8 | 19.6 | 4.22 | 4.22 | 3.15 | 3.15 |
| Stream RAG | 23K | Qwen-OMNI | **34.2** | **37.4** | 5.32 | 3.60 | **2.79** | **1.57** |
| (S. 3.2.2) | 15K | OpusLM | 23.6 | 22.8 | 8.63 | 9.04 | 3.55 | 2.57 |

# 4. Experiment Setup

## 4.1. Evaluation Benchmarks

To evaluate our proposed approach, we construct benchmark datasets featuring spoken queries paired with simulated tool interactions. We build on the CRAG dataset (Yang et al., 2024d) to create *AudioCRAG*, which comprises two variants: **AudioCRAG-Synthetic**, containing 1,862 high-quality spoken queries generated using an in-house text-to-speech (TTS) system, and **AudioCRAG-Human**, consisting of 618 human-recorded queries reused from a curated subset of the WearVox dataset (Lin et al., 2025) and paired with CRAG-style web and KG retrieval to construct a new speech-in/speech-out evaluation setup. Further details on the construction of these benchmarks are in Sec. A.10. We follow the CRAG setup to incorporate both web and KG-based tools, and adopt its robust evaluation methodology, as described in Secs. A.11 and A.12. Additionally, we leverage a random subset of 16,000 questions from the text-based factual question answering dataset TriviaQA (Joshi et al., 2017) to post-train our speech-in, speech-out models, as detailed in Sec. A.14. We also evaluate our approach on another human-spoken benchmark SLUE-SQA (Shon et al., 2023) without any additional fine-tuning, demonstrating that our method transfers robustly across datasets and domains. Importantly, AudioCRAG differs fundamentally from SLUE-SQA in both difficulty and intent. AudioCRAG emphasizes long-tail entities, large-scale web retrieval, and temporal dynamics, leading to substantially lower absolute accuracy across all models. This makes AudioCRAG well suited for analyzing relative performance gains and latency trade-offs, while the widely used SLUE-SQA benchmark provides insight into deployment-ready performance.

## 4.2. Evaluated SOTA Speech-in, Speech-out Models

In this work, we present a comprehensive benchmark of three SOTA speech-in, speech-out conversational systems: Qwen-OMNI (Xu et al., 2025), Kimi-Audio (Ding et al., 2025) and OpusLM (Tian et al., 2025). We evaluate them under both tool-augmented and non-tool-augmented conditions. Further details are provided in S. A.13.

We perform an ablation study (referred to as "Tool Integration" in T. 6) where we post-train the model on sequential query generation (i.e. $P(Q^T|Q)$ in S. 3.1) and output generation, to assess the impact of Stream RAG post-training versus standard post-training on final response generation. Additionally, we conduct an ablation study on open book setting (S. 3.1) using a self-cascade approach with a three-stage inference pipeline: (1) the audio question is used to generate a tool query $Q^T$ (corresponding to the "Query Generation" stage described in S. 3.1); (2) the audio question, and tool results are combined to produce the final text output $X^{res}$ by maximizing $P(X^{res}|Q, R)$; and (3) the audio question, and final text output are used to generate the final speech output $A$ by optimizing $P(A|Q, X^{res})$. Since we teacher-force the text output to obtain the final speech output in stage (3), this self-cascade approach can only be applied to a "thinker-talker" architecture (eq. Qwen-OMNI) or Chain-of-Thought (CoT) style architectures (eg. OpusLM). The motivation for this ablation is to investigate whether the inclusion of RAG references $R$ during the "Response Generation" stage (S. 3.1) affects the quality of the generated speech $A$.

# 5. Results

## 5.1. Impact of Tool Integration and Stream RAG

Table 1 compares Qwen-OMNI, OpusLM, and Kimi Audio across Closed Book, Open Book, and Stream RAG settings on both AudioCRAG-Synthetic (Syn.) and AudioCRAG-

*Table 2.* Exact Match (EM) accuracy and first-token latency for Stream RAG (*Model-Triggered Stream RAG*) with Qwen-OMNI against WavRAG (Chen et al., 2025) and a standard cascaded pipeline. For Stream RAG, the reported latency reflects latency savings relative to utterance end, where negative values indicate that the first token is generated before the end of user speech. WavRAG latencies are lower-bound estimates (see S. A.5).

| Setting | Model | EM | Latency (s) |
|---|---|---|---|
| Cascade | (Whisper→Qwen2.5→VITS) | 36.6 | 1.53 |
| WavRAG | GPT-4o | 40.1 | 0.56 |
| (Chen et al., 2025) | QwenAudio | 30.6 | 0.56 |
| WavRAG-CoT | GPT-4o | 49.2 | 4.34 |
| (Chen et al., 2025) | QwenAudio | 34.0 | 4.34 |
| Stream RAG | Qwen-OMNI | **55.8** | **-0.68** |

**Human (Hum.).** In the Closed Book setting, where models rely solely on their internal knowledge without access to external tools (reference length = 0), all models achieve accuracy scores below 20%. These results highlight the inherent limitations of closed-book approaches in handling complex queries. The Open Book setting, which provides models with access to external information, demonstrates the clear benefits of tool integration. Qwen-OMNI and Kimi Audio's accuracy rises substantially, underscoring the value of leveraging external context. As expected, latency increases due to the additional overhead of retrieving information from external tools.

Notably, our post-training approach to build *Model-Triggered Stream RAG*, as described in S. 3.2.2, delivers significant advancements in both **accuracy and efficiency**. Qwen-OMNI and OpusLM achieve significant accuracy improvements across both benchmarks. To isolate the effect of streaming tool invocation, we compare against a post-trained sequential RAG baseline in Table 6 that uses the same training procedure as Stream RAG but performs retrieval only after the complete user utterance is observed. As discussed in S. 5.6, post-trained sequential RAG achieves 34.9% accuracy, comparable to Stream RAG (34.2%). This indicates that the streaming formulation preserves the accuracy benefits of post-training while reducing user-perceived latency.

While the absolute accuracy scores are modest[1], they are consistent with the evaluation results observed in the CRAG benchmark. Notably, Qwen-OMNI with *Model-Triggered Stream RAG* achieves accuracy comparable to the open book performance of similarly sized LLMs reported in the original CRAG paper (i.e., 34.2 for Qwen-OMNI vs. 32.1 for LLAMA-3 8B Instruct in (Yang et al., 2024d)). Although post-training is performed exclusively on the synthetic dataset, we observe consistent and even greater improvements on the human-spoken benchmark. This setting

---

[1]With text-in/text-out (i.e., perfect ASR and TTS), Qwen-OMNI achieves only 38.9% accuracy, indicating that our method already operates close to the underlying LLM's accuracy ceiling.

also introduces substantial latency savings compared to the Open Book configuration, with Qwen-OMNI and OpusLM achieving 17.2% and 10.6% reductions in tool use latency (i.e. 1 – (Stream RAG Tool-Use Latency / Open-Book Tool-Use Latency) ) on the synthetic benchmark, and even greater savings (i.e. 53.4% and 35.3%) on the human benchmark[2]. While these results demonstrate substantial improvements, S. 5.4 shows that production-grade inference optimization can amplify first-token latency reductions to 57%, reducing response times to 1.36s which is compatible with real-time conversational interaction.

### 5.2. Generalization to SLUE-SQA Benchmark

While AudioCRAG is designed to stress-test long-tail entity retrieval, we next evaluate Stream RAG on SLUE-SQA, a widely used human-spoken QA benchmark that better reflects near-deployment conditions. T. 2 shows that Stream RAG achieves both higher accuracy and substantially lower first-token latency on the SLUE-SQA benchmark. Stream RAG attains an EM score of **55.8**, outperforming all reported WavRAG (Chen et al., 2025) variants[3] and a standard cascaded pipeline consisting of Whisper large-v3 for ASR, Qwen2.5-7B-Instruct for text generation, and a pre-trained VITS model (Hayashi et al., 2020)[4] for speech synthesis. In contrast, Stream RAG achieves a **negative first-token latency of -0.68s** relative to utterance end. A negative latency indicates that response generation begins before the user finishes speaking, enabling more natural, real-time interaction. In practice, this arises because endpoint detection in human-recorded audio often introduces trailing silence, which allows Stream RAG to predict tool queries and initiate generation early. Together, these results demonstrate that Stream RAG effectively balances accuracy and responsiveness on SLUE-SQA, suggesting that its benefits generalize beyond the CRAG benchmark. We conduct a detailed error and efficiency analysis of Stream RAG on SQA and find that **early tool invocation does not correlate with increased errors**: correct and incorrect predictions exhibit nearly identical median latency savings. Latency savings increase with audio duration, while accuracy remains stable across audio durations. Overall, Stream RAG achieves the best accuracy–latency trade-off under moderate early triggering (1–3 s before end of audio), with most remaining failures attributable to query formulation and entity-level

---

[2]Latency calculations on synthetic audio exclude endpoint detection latency, which is required in all production systems. Since stream RAG enables processing without waiting for endpoint detection, including this factor would amplify the observed latency savings, as seen in higher latency savings for human-spoken audio, where endpoint detection errors often introduce trailing silence.

[3]WavRAG retrieves spoken versions of documents, whereas Stream RAG retrieves text passages.

[4]https://huggingface.co/espnet/kan-bayashi_ljspeech_vits

*Table 3.* Results on AudioCRAG-Synthetic for Qwen-OMNI, OpusLM, and Kimi Audio comparing text vs. speech output across Closed Book, Open Book, and *Model-Triggered Stream RAG*.

| Setting | Ref length | Output | Model | Acc. |
|---|---|---|---|---|
| Closed Book | 0 | Text | Qwen-OMNI | 15.0 |
| | 0 | Speech | Qwen-OMNI | 11.1 |
| | 0 | Text | OpusLM | 20.1 |
| | 0 | Speech | OpusLM | 18.4 |
| | 0 | Text | Kimi Audio | 24.2 |
| | 0 | Speech | Kimi Audio | 16.7 |
| Open Book | 23K | Text | Qwen-OMNI | 39.6 |
| (S. 3.1) | 23K | Speech | Qwen-OMNI | 26.3 |
| | 23K | Speech (self-cascade) | Qwen-OMNI | 33.8 |
| | 15K | Text | OpusLM | 26.3 |
| | 15K | Speech | OpusLM | 0.0 |
| | 15K | Speech (self-cascade) | OpusLM | 21.2 |
| | 5K | Text | Kimi Audio | 45.8 |
| | 500 | Speech | Kimi Audio | 21.8 |
| Stream RAG | 23K | Text | Qwen-OMNI | 39.8 |
| (S. 3.2.2) | 23K | Speech | Qwen-OMNI | 34.2 |
| | 15K | Text | OpusLM | 29.7 |
| | 15K | Speech | OpusLM | 23.6 |

reasoning rather than premature retrieval (see T. 12, §A.6).

Beyond single-turn evaluation, we assess Stream RAG in multi-turn conversational scenarios using the CORAL dataset (§A.4). Results show that Stream RAG maintains comparable semantic alignment with ground truth response while achieving 1.83s latency reduction on the second turn, demonstrating that it preserves conversational quality even when user intent depends on preceding dialogue context.

### 5.3. Modality Gap Between Speech and Text Output

A key question is whether Stream RAG exacerbates or mitigates the known modality gap between text and speech output in SDS. Table 3 compares three SOTA models generating text or speech outputs from speech inputs, with and without external tool results. Without tools (Ref length = 0), all models achieve higher accuracy for text output than for speech output. Incorporating tool results generally improves text generation, with Kimi Audio achieving the highest accuracy, while speech output accuracy remains consistently lower across models. The self-cascade approach, in which the model first generates an intermediate text response before producing the final speech output, provides moderate improvements in speech output accuracy for both Qwen-OMNI and OpusLM. These findings underscore a challenge in direct speech generation, particularly when tool results are integrated, as this appears to negatively impact the quality of generated speech responses. Our *Model-Triggered Stream RAG* demonstrates clear advantages: it maintains comparable performance for text output and delivers improvements for speech output, even outperforming the self-cascade approach. However, it still underperforms the text-output baseline, primarily due to difficulties in accurately pronouncing uncommon entity names (see S. A.7).

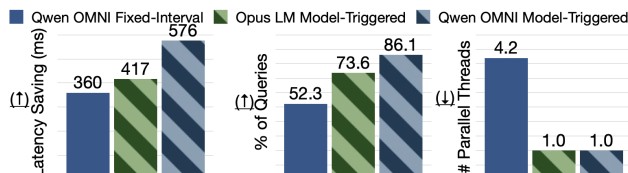

*Figure 4.* Latency savings by Stream RAG approaches (S. 3.2.)

### 5.4. Analysis of Latency Bottlenecks in Tool-Integrated Speech Dialogue with VLLM Integration

The latency reductions in Table 1 represent conservative measurements. To assess real-world deployment performance, we integrated vLLM (Kwon et al., 2023), a production-grade inference engine optimized for throughput and latency, with the Qwen-OMNI speech-to-speech model. This integration yields dramatic improvements. On AudioCRAG-Human, *Model-Triggered Stream RAG* achieves a **57% reduction** in median first-token latency (3.16s → 1.36s), bringing response time close to real-time conversational thresholds. Even on synthetic audio, which lacks endpoint detection delays, we observe a 16.6% reduction (3.50s → 2.92s). These gains are perceptually meaningful, well above the 500ms threshold where users perceive delays as disruptive (see §A.2). Table 4 breaks down these improvements, reporting median (P50) and 90th-percentile (P90, capturing tail latency) timings decomposed into tool-use and response-generation components[5]. Assuming parallel tool access, the reported latency corresponds to the maximum across all tools. The majority of latency arises from web retrieval, which our streaming approach successfully masks. Notably, tool-results generation time decreases from 2.78s to 2.20s, enabling earlier response synthesis.

### 5.5. Human Evaluation

While automatic metrics provide useful insights into factual correctness and latency, they do not fully capture the user-perceived quality of speech dialogue systems. We therefore conduct a human evaluation comparing Stream RAG against a strong cascaded baseline (Whisper → Qwen2.5 → VITS) on the SLUE-SQA benchmark.

We randomly sample 100 test utterances and collect two independent ratings per sample (200 ratings total). Annotators evaluate: ① **Answer Correctness**: How accurately the response answers the user query compared to the reference answer (5 = perfectly correct, 1 = completely incorrect). ② **Perceived Latency**: How responsive the system feels before beginning its answer (5 = instant response, 1 = very long delay). In addition, annotators provide an overall pairwise preference between the two systems.

For overall preference, annotators preferred Stream RAG in 74.5% of ratings compared to 10.0% for the cascaded base-

---

[5]Baseline results without vLLM optimization in Table 7

*Table 4.* First Token Latency breakdown, showing median (P50) and 90th percentile (P90) timings, for the Qwen-OMNI on AudioCRAG-Synthetic after integrating with vLLM.

| Model | Setting | P | Latency (sec) | | | |
|---|---|---|---|---|---|---|
| | | | Tool Use Latency | | Response Gen | Total |
| | | | Query Gen | Tool Results Gen | | |
| Qwen-OMNI | Open Book (S. 3.1) | P50 | 0.05 | 2.78 | 0.67 | 3.50 |
| | | P90 | 0.07 | 4.90 | 0.86 | 5.83 |
| | Stream RAG (S. 3.2.2) | P50 | 0.05 | 2.20 | 0.67 | 2.92 |
| | | P90 | 0.07 | 4.37 | 0.86 | 5.30 |

*Table 5.* Human evaluation on 100 SLUE-SQA samples. Higher is better.

| Metric | Stream RAG | Cascade |
|---|---|---|
| Correctness (↑) | **3.75** | 2.71 |
| Perceived Latency (↑) | **4.27** | 3.15 |

*Table 6.* Comparison under different post-training conditions

| Post-Training | Ref length | Model | Acc. |
|---|---|---|---|
| Tool Integration (S. 4.2) | 15K | OpusLM | 22.4 |
| Stream RAG (S. 3.2.2) | 15K | OpusLM | 23.6 |
| Tool Integration (S. 4.2) | 23K | Qwen-OMNI | 34.9 |
| Stream RAG (S. 3.2.2) | 23K | Qwen-OMNI | 34.2 |

line, while 15.5% of ratings indicated no preference. Inter-annotator agreement was 77%. Notably, Table 5 shows that improvements are observed for both correctness (+1.04 absolute) and perceived latency (+1.12 absolute), suggesting that the latency savings do not come at the expense of response quality. These results demonstrate that the latency reductions achieved by Stream RAG translate into a noticeably better user experience while simultaneously improving answer correctness.

### 5.6. Ablation Study

**Stream RAG approaches**: Figure 4 highlights the substantial latency savings enabled by the *Model-Triggered Stream RAG* (S. 3.2.2), compared to the *Fixed-Interval Stream RAG* (S. 3.2.1). Three key metrics are evaluated: overall latency savings, the percentage of queries benefiting from reduced latency, and the number of parallel threads required. Even without any post-training, the *Fixed-Interval Stream RAG* approach already reduces tool usage latency (3.37s in T. 7) by 10.7% for Open Book Qwen-OMNI, demonstrating its plug-and-play compatibility with any existing SDS. The *Model-Triggered Stream RAG* method, utilizing Qwen-OMNI, consistently delivers superior performance. It achieves greater average latency reductions and benefits a higher proportion of queries with improved response times. *Model-Triggered Stream RAG* requires only a single parallel thread, representing an advancement in resource efficiency compared to the *Fixed-Interval Stream RAG* approach. We provide a detailed analysis of tool call frequency and its relationship to utterance duration in Appendix A.16.

**Post-training Strategies**: Table 6 presents performance comparison under different post-training conditions. Incorporating streaming tool query generation during post-training (S. 3.2.2) results in comparable performance for both models. These results suggest that *Model-Triggered Stream RAG* can be integrated into post-training process without negatively impacting model performance.

## 6. Conclusion and Discussion

We introduce the first approach for streaming tool-query scheduling in E2E SDS, enabling tool calls to be issued while the user is still speaking. Our *Model-Triggered Stream RAG* approach enhances the model's ability to leverage retrieved information and autonomously decide when to trigger new tool queries, resulting in improved accuracy and responsiveness. Experiments show that tool integration increases factual QA accuracy by 20% absolute over closed-book models, while reducing first-token latency by up to 57% over open-book models, preserving natural conversational flow. We envision Stream RAG as a latency-critical layer that complements ongoing advances in speech modeling, enabling immediate responsiveness as underlying model accuracy continues to improve. Future work includes extending Stream RAG to more complex multi-hop and multi-query retrieval settings, where the system may need to iteratively refine retrieval plans over longer conversations while maintaining low latency and response quality.

## Impact Statement

In this work, we propose an approach for integrating external tool usage directly into E2E speech-in, speech-out dialogue systems, and as such do not see any new ethical concerns arising as a result of our work. We are dedicated to upholding the highest standards of research ethics and reproducibility. All experiments utilize open-source models, ensuring no privacy violations, and we will release all code to the public to facilitate transparency and further research in the community. To ensure the reproducibility of our work, we provide comprehensive details throughout the paper and supplementary materials. The prompts used for query and response generation are included in S. A.18, while the evaluation prompts for the LLM-as-judge setup are detailed in

S. A.19. Further information regarding tool usage and our experimental setup can be found in S. A.11 and A.13.The hyperparameters for our *Model-Triggered Stream RAG* post-training are summarized in Tables 19–21.

The AudioCRAG-Human dataset reuses human-recorded audio from the WearVox dataset (Lin et al., 2025), which was collected from consenting adult participants. All participants provided informed consent prior to their involvement in the study. Approximately 100 participants, all over the age of 18, were recruited by a third-party vendor and compensated for their participation. Personal identifying information was either obfuscated or not collected. The dataset is intended for evaluation purposes only and must not be used for training.

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

*Table 7.* First Token Latency breakdown, showing median (P50) and 90th percentile (P90) timings, for the Qwen-OMNI on AudioCRAG-Synthetic.

| Model | Setting | P | Latency (sec) | | | |
| | | | Tool Use Latency | | Response Gen | Total |
| | | | Query Gen | Tool Results Gen | | |
| Qwen-OMNI | Open Book (S. 3.1) | P50 | 0.59 | 2.78 | 2.52 | 5.90 |
| | | P90 | 0.85 | 4.90 | 3.25 | 9.00 |
| | Stream RAG (S. 3.2.2) | P50 | 0.59 | 2.20 | 2.52 | 5.32 |
| | | P90 | 0.85 | 4.37 | 3.25 | 8.47 |

# A. Appendix

### A.1. Analysis of Latency Bottlenecks in Tool-Integrated Speech Dialogue without vLLM Integration

Table 7 provides a comprehensive breakdown of latency measurements for the Qwen-OMNI speech-to-speech model, evaluated in both the Open Book setting and our proposed *Model-Triggered Stream RAG* setting on AudioCRAG-Synthetic. We report both median (P50) and 90th percentile (P90) values for each stage of the processing pipeline. The latency is decomposed into three main components: tool query generation, tool result generation, and speech response synthesis, with measurements provided for both the first token outputs (We also provide latency measurements for last token output in T. 17). For both tool query and tool result generation, it is assumed that all tools are accessed in parallel; thus, the reported latency corresponds to the maximum query or result generation time among all tools. The majority of this latency arises from leveraging external web pages, which introduces significant delays, most notably, increasing the first token latency by 2.3x in Open Book Setting. Notably, our *Model-Triggered Stream RAG* setting enables early generation of tool results, successfully reducing P50 first token latency by 9.8% and tool use latency by 17.2%.

### A.2. Human Perception of Latency Savings

Prior work on human conversational timing (Jacoby et al., 2024; Stivers et al., 2009) shows that the average turn-taking gap in natural human dialogue is approximately 239 ms, with delays beyond 500 ms perceived as unnatural. Complementary industry analyses (Vapi, 2025) similarly report that speech latency exceeding 500 ms begins to degrade user experience, causing users to interrupt or disengage. In this context, our observed latency reductions, particularly a 1.8 s improvement (T. 1) for human-spoken audio, represent a substantial enhancement relative to perceptual thresholds in human conversation. Moreover, our integration of efficient inference backends (e.g., vLLM for Qwen-OMNI) achieves P-50 latency near 670 ms, comparable to production-quality voice AI systems that target sub-second responsiveness. These results indicate that our latency reductions are perceptually significant, contributing directly to smoother conversational turn-taking and more natural user experience in speech-to-speech dialogue systems.

### A.3. Evaluating Robustness to Misfired Early Tool Calls

We further evaluate the robustness of Stream RAG under adversarial intermediate conditions on the SLUE-SQA dataset. In this stress test, Stream RAG achieves a recovery accuracy of **65.4%**, demonstrating that the model can reliably correct erroneous intermediate tool signals and recover the correct final retrieval target in the majority of cases. Concretely, we construct an adversarial setting by replacing the intermediate query $\hat{Q}_b^{\text{prev}}$ with a misleading query $Q_b^{\text{neg}}$, where $Q_b^{\text{neg}}$ is sampled from the set of all final tool queries generated by Stream RAG across the SLUE-SQA dataset, excluding the query corresponding to the current example. We define the reference query as the final tool query produced under the standard (non-adversarial) setting,

$$\hat{Q}_b^T = \arg\max_{Q_b^T} P(Q_b^T \mid Q, \hat{Q}_b^{\text{prev}}),$$

and evaluate the perturbed prediction

$$\hat{Q}_b^{\text{perturb}} = \arg\max_{Q_b^T} P(Q_b^T \mid Q, Q_b^{\text{neg}}).$$

A perturbed prediction is considered correct if it either exactly matches $\hat{Q}_b^T$ or retrieves the same top-$k$ document as $\hat{Q}_b^T$ with $k \leq 5$. These results indicate that Stream RAG is robust to incorrect intermediate tool context and can dynamically revise tool usage decisions as additional user speech becomes available, without reliance on external verification mechanisms.

*Table 8.* Performance comparison of ROUGE-L, ROUGE-L First-25 (ROUGE-L computed over the first 25 words of the generated response), and first-token latency for Qwen-OMNI in the speech-in/speech-out setting. Results are reported on the second turn of two-turn interactions in the CORAL dataset. The reported latency reflects latency savings relative to utterance end, where negative values indicate that the first token is generated before the end of user speech.

| Setting | Model | ROUGE-L | ROUGE-L First-25 | First Token Latency) |
|---|---|---|---|---|
| Text-in Text-Out (Cheng et al., 2025) | Qwen2.5-7B | 13.1 | ✗ | ✗ |
| Open Book | Qwen-OMNI | 13.6 | 10.9 | 1.01 |
| Stream RAG | Qwen-OMNI | 12.9 | 11.5 | - 0.82 |

*Table 9.* Qualitative error cases illustrating the modality gap in speech-in/speech-out systems. While reference answers contain correct entity names, direct speech generation can produce phonetic distortions for named entities, despite correct underlying intent.

| Question | Reference Answer | Speech-to-Speech Output (Whisper Transcript) |
|---|---|---|
| Who are the members of the band The 1975? | The members of The 1975 are Matty Healy, Adam Hann, Ross MacDonald, and George Daniel. | the members of the one1995 are mon healy har han ross mcdonald and jor daniel |
| Who has more goals, Ronaldo or Messi? | Ronaldo holds the record for most official goals in a career. | ronald to. if you have any other questions feel free to ask. |

## A.4. Evaluating Performance in Multi Turn Scenarios

Table 8 evaluates Stream RAG in a two-turn conversational setting on the CORAL dataset, highlighting its ability to deliver substantial latency reductions while maintaining strong response quality. On the second turn, Stream RAG achieves a **1.83 s reduction in first-token latency** relative to the Open-Book baseline, demonstrating that the latency benefits observed in single-turn settings extend naturally to multi-turn dialogue.

In terms of response quality, Stream RAG exhibits a small decrease in full ROUGE-L compared to Open-Book (12.9 vs. 13.6), which is largely attributable to differences in response length: Open-Book often produces longer and more elaborative outputs, whereas Stream RAG generates more concise and focused answers. This effect is reflected in ROUGE-L First-25, which measures early semantic alignment and shows a slight advantage for Stream RAG (11.5 vs. 10.9), indicating that Stream RAG's concise responses achieve equal or better early semantic alignment.

We evaluate on 470 two-turn interactions, where both turns are synthesized using OpusLM (Tian et al., 2025) and transcribed with Whisper (Radford et al., 2023). To ensure transcription fidelity, we automatically filter the data by retaining only examples with no intelligibility errors, defined as zero word error rate (WER) between the Whisper hypothesis and the reference transcription. Identical dialogue histories are used for both methods. Taken together, these results demonstrate that Stream RAG enables *simultaneous gains in responsiveness and early semantic accuracy* in multi-turn spoken dialogue, substantially reducing user-perceived latency without sacrificing the quality of the generated response.

## A.5. Latency Computation for WavRAG

To facilitate a fair latency comparison with WavRAG (Chen et al., 2025), we report a best-case estimate of its first-token latency, as E2E latency measurements are not provided in the original work. We approximate two components: (1) retrieval latency, using a lightweight text-based pipeline consisting of a web query followed by a BGE reranker to retrieve the top-3 text passages, which likely underestimates the cost of spoken-document retrieval used in WavRAG; and (2) generation latency, using measured Qwen-OMNI decoding latency with vLLM optimization. For WavRAG-CoT, we add the instruction "Please think step by step and explain your answer." to the prompt as recommended in the WavRAG paper. Together, these approximations provide a best-case reference for WavRAG latency against which we compare Stream RAG.

## A.6. Analysis of Stream RAG failure Cases

We further analyze the success and failure cases of Stream RAG on the SQA dataset. We examine first-token latency for both correctly and incorrectly answered queries and observe nearly identical latency savings across the two groups. In particular, queries answered with exact match achieve a median first-token latency of -0.70 s, while incorrectly predicted queries

*Table 10.* Exact-match accuracy as a function of latency saving relative to the Open-Book setting, where retrieval is performed only after endpoint detection.

| Latency Saving (s) | EM (%) | Examples (%) |
|---|---|---|
| 0 s | 52.2 | 9.7 |
| 0–1 s | 55.1 | 33.1 |
| 1–3 s | **58.2** | 39.6 |
| >3 s | 53.1 | 17.6 |

*Table 11.* Exact-match accuracy and median latency savings as a function of audio duration. Latency savings increase with utterance length, while accuracy remains stable across all duration ranges.

| Audio Duration | Latency Saving (s) | EM (%) | Examples (%) |
|---|---|---|---|
| 0–3 s | 0.74 | 56.8 | 4.0 |
| 3–5 s | 1.15 | 53.9 | 36.4 |
| 5–7 s | 1.60 | 57.0 | 31.3 |
| 7–10 s | 2.39 | 55.8 | 20.1 |
| >10 s | 4.40 | 57.3 | 8.1 |

achieve -0.66 s. This indicates that early tool invocation does not disproportionately contribute to incorrect predictions.

We analyze the relationship between accuracy and latency reduction by grouping examples according to the amount of latency saved relative to the *Open-Book* setting, in which retrieval is triggered only after endpoint detection of the user's utterance. Table 10 reports exact-match (EM) accuracy across different latency-saving regimes. First, the highest accuracy is achieved in the moderate early-triggering regime (1–3 s latency saving). Second, even aggressive early triggering (>3 s saved) maintains competitive performance, achieving over 53% EM. Finally, accuracy does not decrease monotonically with increased latency savings, indicating that earlier retrieval does not inherently degrade answer quality. Moderate early triggering provides the best balance between responsiveness and correctness, while more aggressive triggering incurs only a minor accuracy drop. Overall, these results suggest that *Stream RAG* is robust to early commitment, and that most remaining errors stem from query formulation or answer selection rather than premature tool invocation.

Table 11 analyzes the effect of audio duration on both accuracy and responsiveness. As expected, longer utterances provide greater opportunities for early tool invocation, resulting in progressively larger latency savings—from 0.74 s for short (0–3 s) queries to 4.40 s for utterances longer than 10 s. Importantly, exact-match accuracy remains stable across all duration ranges (53.9–57.3%), indicating that Stream RAG maintains consistent answer quality regardless of input length. These results suggest that streaming retrieval scales favorably with utterance duration, improving responsiveness without introducing a trade-off in accuracy.

Table 12 further illustrates the strengths and limitations of Stream RAG through representative examples. In the correct high-latency-saving case, the question about U.S. states with Pacific coastlines contains the key disambiguating concept ("Pacific coastlines") early in the utterance, allowing the model to confidently issue a retrieval query and correctly identify the remaining state. In contrast, the incorrect high-latency-saving example involving the Venetian painter Jacopo Robusti (Tintoretto) fails due to entity confusion: although the model retrieves relevant context, it conflates the artist with his teacher Titian. This suggests that richer query formulation, such as explicitly incorporating the listed painting titles, could help disambiguate closely related entities in such cases. Overall, these examples reinforce that early tool invocation is effective when salient entities appear early.

In the low latency-saving regime, both correct and incorrect predictions occur despite conservative triggering, suggesting that delayed retrieval alone does not guarantee improved accuracy. Together, these examples highlight that early tool invocation is not inherently responsible for errors. Instead, prediction quality is primarily governed by query formulation and downstream model reasoning.

*Table 12.* Qualitative examples illustrating the interaction between latency savings and prediction correctness.

| | Correct | Incorrect |
|---|---|---|
| **High Latency Saving ($\geq 3$ s)** | *Question:* Three of the contiguous states of the USA have Pacific coastlines. Washington and California are two. Which is the third?

*Predicted Query:* Contiguous US states with Pacific coastlines | *Question:* Which 16th century Venetian, born Jacopo Robusti, studied under Titian and painted *St George and the Dragon*, *Belshazzar's Feast*, *The Last Supper*, and *Paradise*?
*Predicted Query:* 16th century Venetian painter Jacopo Robusti Titian Saint George painting |
| **Low Latency Saving ($\leq 1$ s)** | *Question:* What did Luther fear to cause him to avoid marriage?
*Predicted Query:* Luther fear of marriage | *Question:* When was the last Super Bowl in California?
*Predicted Query:* Super Bowl in California last date |

## A.7. Qualitative Errors with Speech Output

To better understand the persistent accuracy gap between speech-in/text-out and speech-in/speech-out systems, we analyze representative qualitative error cases. As shown in Table 9, direct speech generation frequently struggles with the accurate pronunciation of uncommon nouns and named entities, even when the underlying semantic content is correct. While speech-to-text outputs preserve entity names such as *The 1975*, *Matty Healy*, and *Ronaldo*, the corresponding speech-to-speech outputs exhibit phonetic distortions (e.g., "one1995", "mon healy", "ronald to"). These errors highlight a modality gap between text and speech generation, particularly for entity-heavy answers retrieved via external tools, and help explain why speech outputs lag behind text outputs despite strong text-generation performance.

These findings suggest two concrete directions for future work. First, speech-in/speech-out systems could benefit from more entity-aware speech generation, such as incorporating pronunciation lexicons for rare names or retrieval-guided phoneme generation when external tools introduce unfamiliar entities. Second, adopting speech-aware evaluation, for example, AudioLM-style judges, may reduce over-penalization of valid spoken outputs that are difficult for pre-trained ASR systems (e.g., Whisper) to transcribe perfectly. Together, these directions offer promising avenues for narrowing the remaining gap between speech and text outputs.

## A.8. Benchmarking Text Dialogue Systems for Tool Usage

Recent work has shown that large language models can be augmented with external tools, such as web search, APIs, databases, and function calls, to overcome knowledge staleness and improve factual reliability. Early systems such as WebGPT (Nakano et al., 2022) and MRKL (Karpas et al., 2022) formalized the idea of LLMs acting as controllers over external modules, while ReAct (Yao et al., 2023) introduced interleaved reasoning and acting to decide when to invoke tools during generation. Subsequent efforts have focused on learning tool invocation policies directly: Toolformer (Schick et al., 2023) demonstrated self-supervised training for deciding when and how to call tools, while later systems such as Gorilla (Patil et al., 2024) and NexusRaven (team, 2023) expanded this paradigm to large API spaces with improved function-call accuracy. These approaches primarily operate in text-based settings and assume synchronous, turn-based interaction, where tool calls are issued after the full user query is available.

Recent advances in benchmarking text-based dialogue systems for tool usage (Chen et al., 2023b; Ouyang et al., 2025; Cheng & Dou, 2025; Cohen et al., 2025; Xiong et al., 2024b; Vu et al., 2023; Xiong et al., 2024a; Peng et al., 2024; Su et al., 2025; Ni et al., 2025; Li et al., 2023; Qin et al., 2023; Guo et al., 2024; Trivedi et al., 2024; Patil et al., 2025; Yang et al., 2024c) have primarily focused on evaluating factual question answering and task completion within simulated environments. The CRAG benchmark (Yang et al., 2024d) is a leading example, featuring 4,409 question-answer pairs and providing mock APIs for both web and knowledge graph (KG) search. CRAG supports a range of KG and web retrieval tasks, and highlights key challenges such as hallucinations in retrieval-augmented generation (RAG) and the importance of leveraging KGs and search ranking to improve factual accuracy. Evaluation is conducted automatically using two LLM judges. SimpleQA (Wei et al., 2024) is another widely adopted benchmark, designed to assess language models on short, fact-seeking questions. With 4,326 adversarially collected questions spanning diverse topics and a straightforward grading scheme based on single, indisputable answers, SimpleQA provides a robust testbed for factual accuracy. Moving

beyond question answering, WebArena (Zhou et al., 2024) offers a simulated environment for evaluating dialogue agents on web-based tasks using fully functional websites, enabling assessment of more complex, action-oriented behaviors. While these benchmarks have significantly advanced the evaluation of tool-augmented dialogue systems, they remain largely limited to text-based interactions and do not fully address the unique challenges presented by speech-in, speech-out systems.

### A.9. Multimodal Benchmarks for Tool Usage

Recent benchmarks (Mei et al., 2025; Yu et al., 2025; Luo et al., 2024) have extended tool-augmented dialogue evaluation to multimodal and longer-context scenarios. The m&m's benchmark (Ma et al., 2024) evaluates LLMs on multi-step, multi-modal tasks using a diverse set of 33 tools, including public APIs and multimodal models such as off-the-shelf automatic speech recognition (ASR) models, highlighting the potential for developing agents that leverage audio-based tools. CRAG_MM (Meta CRAG-MM Challenge Organizers, 2025) builds on the original CRAG benchmark by introducing visual question answering (QA) tasks that combine images and text-based queries, utilizing mock APIs for both image descriptions and web search. For video understanding and long-context reasoning, the Video Web Arena (Jang et al., 2025) benchmark evaluates multimodal agents on tasks involving 2,021 manually crafted tutorial videos. While these benchmarks advance the field by incorporating multimodal tools, they still do not evaluate systems in speech-in, speech-out scenarios. Related work has also begun to explore RAG beyond text, such as RAG for text-to-music generation (Gonzales & Rudzicz, 2024), but remains outside the scope of spoken dialogue interaction.

### A.10. Audio CRAG Benchmark

We begin with the CRAG dataset (Yang et al., 2024d), which contains 2,706 text queries. Since these queries are not directly suitable for speech-based evaluation, we first identify those requiring adaptation before TTS conversion. Through careful manual inspection, we determine that queries containing dates or brackets benefit from rewriting to ensure naturalness and clarity in spoken form. In total, we identify 569 such queries and rewrite them using a large language model (LLAMA-4 Maverick). The resulting 569 rewritten queries, combined with the remaining original queries, form the basis for our spoken version of CRAG, which we term *AudioCRAG*. We follow the CRAG setup to incorporate web and KG-based tools and adopt its robust evaluation setup, as detailed in S. A.11 and A.12.

**Audio CRAG Synthetic**: To generate spoken queries, we process all 2,706 queries through our in-house TTS system. We then apply a rigorous filtering procedure to remove queries with intelligibility issues, specifically excluding any utterances for which Whisper (Radford et al., 2023) hypotheses exhibit a non-zero word error rate. We also remove utterances with suboptimal audio quality, as determined by UTMOS (Saeki et al., 2022) scores below 3.5. This results in a high-quality set of 1,862 spoken queries, which we refer to as the *AudioCRAG-Synthetic* benchmark.

**Audio CRAG Human**: To further enhance the realism and diversity of our evaluation, we introduce the AudioCRAG-Human benchmark, which consists of 618 human-recorded spoken queries. We reuse high-quality audio recordings from the WearVox dataset, originally collected for speech-in/text-out evaluation, by selecting a subset of recordings and constructing a new speech-in/speech-out evaluation setup aligned with CRAG-style retrieval over web pages and knowledge graph. The recordings originate from a diverse pool of participants, capturing natural variations in speech, accent, and prosody. By repurposing these recordings within a distinct evaluation framework, AudioCRAG-Human enables a more comprehensive assessment of speech-in/speech-out systems under realistic conditions, offering valuable insights into robustness, generalization, and the effectiveness of tool integration beyond synthetic speech.

### A.11. Tool Usage Setup

**Audio CRAG**: To enable effective tool usage, we build upon the CRAG framework by integrating two complementary information sources: web search, which provides access to fresh and dynamic content, and a knowledge graph, which offers structured and reliable information. For web search, we aggregate all 100,000 documents from the CRAG corpus and employ a BGE-based re-ranker[6] (Xiao et al., 2023) to index and retrieve the top 50 most relevant documents for each query. These documents are then segmented into chunks and re-ranked using the same BGE model based on their similarity to the query, ensuring highly contextually relevant retrieval. Meanwhile, queries to the knowledge graph are performed via a simulated API, adhering to the methodology established in CRAG [7].

---

[6]https://huggingface.co/BAAI/bge-large-en-v1.5
[7]https://github.com/facebookresearch/CRAG/tree/main/mock_api

**SLUE-SQA** (Shon et al., 2023): We follow the evaluation protocol of WavRAG (Chen et al., 2025) as closely as possible. We index all 1,969 passages from the SLUE-SQA test set and use the generated web query to retrieve the **top-3** most relevant passages with a BGE reranker. Unlike WavRAG, which retrieves spoken document audio, our setup retrieves the corresponding text passages, reflecting a text-based retrieval backend while keeping the overall retrieval pipeline comparable.

**CORAL** (Cheng et al., 2025): We follow the evaluation setup of (Cheng et al., 2025) as closely as possible. We index all 201K passages in the CORAL passage corpus and use the generated web query to retrieve the top-50 passages with a BGE reranker. To generate the web query, we condition on the audio from both the first and second dialogue turns.

We use the same tool-query generation prompt across all three datasets to ensure a consistent comparison. The prompt is provided in Section A.18.

*Table 13.* Results on AudioCRAG-Synthetic for Qwen-OMNI, OpusLM, and Kimi Audio comparing text vs. speech output across Open Book setting showing average rates of accurate, hallucinated, and missing responses, as well as overall truthfulness scores for each system.

| Ref length | Output | Model | Score | Acc. | Halluc | Miss. |
|---|---|---|---|---|---|---|
| 0 | Text | Qwen-OMNI | -13.1 | 15.0 | 28.1 | 56.9 |
| 0 | Speech | Qwen-OMNI | -21.1 | 11.1 | 32.3 | 56.6 |
| 0 | Text | OpusLM | -44.3 | 20.1 | 64.3 | 15.6 |
| 0 | Speech | OpusLM | -47.9 | 18.4 | 66.2 | 15.4 |
| 0 | Text | Kimi Audio | -38.5 | 24.2 | 62.7 | 13.1 |
| 0 | Speech | Kimi Audio | -53.5 | 16.7 | 70.3 | 12.9 |

### A.12. Evaluation Setting

**AudioCRAG**: Similar to previous work (Yang et al., 2024d), we employ model-based automatic evaluation. We use a three-way scoring system, assigning scores of 1, -1, and 0 for accurate, incorrect, and missing answers, respectively. The evaluation is conducted using the Llama 4-maverick LLM evaluator. For speech outputs, we first transcribe the audio using Whisper large v3 (Radford et al., 2023) before passing the transcriptions to the LLM evaluator. In this study, our primary focus is on enhancing system accuracy; therefore, we report average accuracy values in Tables 1 and 3. For additional context, we also provide the average rates of accurate, hallucinated, and missing responses, as well as overall truthfulness scores for each system in the Open Book Setting (see Table 13). Notably, our results indicate that Qwen-OMNI was less likely to generate hallucinated responses compared to OpusLM and Kimi Audio.

**SLUE-SQA**: We follow the standard evaluation protocol from prior work (Chen et al., 2025) and report Exact Match (EM) accuracy, a binary metric that assigns a score of 1 if the ground-truth answer string appears in the generated response and 0 otherwise.

**CORAL**: Consistent with the original CORAL benchmark (Cheng et al., 2025), we evaluate response quality by comparing generated responses with reference answers using the rule-based ROUGE-L metric (Lin, 2004).

**Latency Metrics**: We report the following latency measures throughout the paper. First-token latency is defined as the time until the first synthesized audio token is generated. Tool-use latency corresponds to the time spent on external tool invocation, including both tool-query generation by the model and retrieval of results from the tool. Negative first-token latency indicates that response generation begins before the user finishes speaking, reflecting effective overlap between response generation and incoming speech.

### A.13. Experiment Setup of SOTA speech-in, speech-out models

Several recent works (Nguyen et al., 2023; Meng et al., 2024; Zhang et al., 2024; Arora et al., 2025b) have introduced audio foundation models for E2E SDS. In this work, we experiment with the following three speech-in, speech-out dialogue systems.

**Qwen-OMNI** (Xu et al., 2025) (Qwen2.5-Omni-7B) is a E2E multimodal model that seamlessly integrates diverse input modalities—including text, images, audio, and video—and generates both text and natural speech responses in a real-time streaming fashion. It leverages an innovative Thinker-Talker architecture, where the Thinker module performs high-level reasoning to produce a text response, which is then used by the Talker module, conditioning on both the text and the

Thinker's hidden representations, to generate streaming speech output.

**OpusLM** (Tian et al., 2025) is an open-source speech-in, speech-out model post-trained to directly answer complex semantic and factual questions from raw audio inputs, through Chain-of-Thought reasoning.

**Kimi Audio** (Ding et al., 2025) is a universal audio foundation model that unifies audio understanding, generation, and conversational abilities within a single framework. Pre-trained on over 13 million hours of diverse audio and text data, Kimi Audio achieves state-of-the-art performance across a wide range of audio benchmarks, including audio understanding and speech conversation tasks.

For tool-augmented scenarios, retrieval results are provided up to each model's maximum token limit ("Ref length" in Tables), maintaining a 2:1 ratio of web page to KG results. Specifically, we observe that Kimi-Audio is currently optimized for handling tool result references up to a certain length. When this limit is exceeded, an error arises during the audio detokenization process, specifically within the rotary embedding mechanism, highlighting an architectural constraint in processing longer input sequences or larger reference contexts. Addressing this limitation presents a valuable opportunity for future model enhancements.

Our evaluation encompasses both speech-in-text-out and speech-in-speech-out scenarios. For the Fixed-Interval Stream RAG setting (Section 3.2.1), intermediate tool queries are generated at consistent 1-second intervals. In the Model-Triggered Stream RAG setting, the model dynamically determines the need for a tool call after processing each 500ms block. This approach allows us to utilize a smaller chunk size, as only a single tool call thread is required for Model-Triggered Stream RAG, thereby enabling more efficient and responsive processing.

### A.14. Post-Training Data Preparation

This subsection details the experimental setup for post-training the pretrained speech-in, speech-out model to significantly enhance its tool usage capabilities, as outlined in S. 3.2.2. We leverage a random subset of 16,000 questions from the text-based factual question answering dataset TriviaQA (Joshi et al., 2017), which contains 97,000 questions and 662,659 associated web documents. For *Model-Triggered Stream RAG*, we further compute word-level timestamps using a pre-trained ASR model, OWSM CTC v4 1B (Peng et al., 2025), enabling us to generate partial ASR transcripts $X_b^{asr}$ at 500 ms intervals. Note that if word occurs at boundary of block $b$, it is excluded from $X_b^{asr}$. For each partial transcript $X_b^{asr}$, we generate corresponding pseudo ground truth queries $\overline{Q_b^T}$ using LLAMA-4-Maverick to simulate incremental user input. To simulate realistic tool usage, we concatenate all documents and employ a web query reranker to retrieve the top 50 most relevant documents for each query. The text questions from this 16k subset are converted into discrete speech tokens using text-to-speech synthesis with the corresponding pretrained speech-in, speech-out model. Recognizing that TriviaQA answers are typically single named entities, we further transform the queries into a conversational style using LLAMA-4-Maverick, making them more suitable for dialogue-based evaluation.

### A.15. Ablation Results for Negative Sampling Strategy

*Table 14.* Ablation Results for Negative Sampling Strategy in *Model-Triggered Stream RAG* (S. 3.2.2).

| Scenario | Input | Ref Length | Post Train Data | Output | Model | Acc. |
|---|---|---|---|---|---|---|
| Open Book | Speech | 15K | 0 | Text | Qwen-OMNI | 39.6 |
| Post-train (S. 3.2.2) | Speech | 15K | 16K | Text | Qwen-OMNI | 39.8 |
| - Negative sampling | Speech | 15K | 16K | Text | Qwen-OMNI | 36.5 |

This table presents the ablation results evaluating the impact of deliberately injecting negative samples during post-training (Eq. 5). The findings highlight that, without negative sampling, streaming tool query generation can reduce final accuracy in text output settings, primarily due to errors in final query generation, as detailed in S. 3.2.2. In contrast, our negative sampling approach significantly enhances the model's robustness, enabling it to recover from intermediate prediction errors. This leads to consistently high accuracy while also achieving notable latency reductions (Tab. 7).

*Table 15.* Example KG Queries, and Web Queries generated by Qwen-OMNI in Open Book setting

| ASR Transcript of Question $X^{\text{asr}}$ | Web Query $\hat{Q}^{\text{web}}$ | KG Query $\hat{Q}^{\text{KG}}$ |
|---|---|---|
| which of nolan greenwald's movies has achieved the highest level of box office success on a global scale? | Nolan Greenwald's highest-grossing movie | {'domain': 'movie', 'movie_name': "Nolan Greenwald's movies", 'movie_aspect': 'revenue'} |
| who has played drums for the red hot chili peppers? | Red Hot Chili Peppers drummers | {'domain': 'music', 'artist_name': 'Red Hot Chili Peppers', 'artist_aspect': 'member'} |
| what's the current stock price of tortoise midstream energy fund? | Tortoise Midstream Energy Fund stock price | {'domain': 'finance', 'market_identifier': 'Tortoise Midstream Energy Fund', 'metric': 'price', 'datetime': '02/28/2024'} |
| what was the volume of trading in cabot corporation's stock on the most recent day that dividends were distributed? | CABOT Corp stock trading volume on dividend distribution date | {'domain': 'finance', 'market_identifier': 'Cabot Corporation', 'metric': 'dividend', 'datetime': '02/28/2024'} |
| which movie won the academy award for best film in 2020? | 2020 Academy Award for Best Picture | {'domain': 'movie', 'movie_aspect': 'oscar_awards', 'year': 2020} |
| which teams have won against phoenix suns during 2022-12? | Teams that beat Phoenix Suns in December 2022 | {'domain': 'sports', 'sport_type': 'basketball', 'team': 'Phoenix Suns', 'datetime': '2022-12-15'} |

### A.16. Tool Call Frequency Analysis

A natural question is whether Stream RAG increases the total number of tool calls compared to traditional retrieve-after-endpoint RAG. The answer is yes; however, this is an intentional trade-off for reducing user-perceived latency.

For Fixed-Interval Stream RAG (S. 3.2.1), tool queries are generated at predefined intervals during user speech, resulting in multiple retrieval requests per utterance. These requests are executed in parallel rather than sequentially, allowing retrieval latency to be overlapped with ongoing speech. Furthermore, the Reflector module selects the earliest sufficient retrieval result for response generation, while later redundant retrievals can be discarded.

For Model-Triggered Stream RAG (S. 3.2.2), the model autonomously determines whether a new query should be issued at each interval. Rather than triggering retrieval at every step, the model learns to refine tool queries only when additional speech provides new information. Importantly, only a single retrieval thread remains active at any time: whenever a new query is generated, any ongoing retrieval associated with the previous query is immediately terminated. Consequently, Model-Triggered Stream RAG behaves more like online query refinement than unconstrained parallel search.

Table 18 reports the average number of tool calls issued by Model-Triggered Stream RAG on SLUE-SQA. Compared to traditional RAG, which issues a single retrieval request after the utterance endpoint, Stream RAG generates an average of 4.7 tool calls per utterance. As expected, longer utterances require more query refinements, leading to a gradual increase in tool call frequency.

These results indicate that Stream RAG achieves latency reduction through a moderate increase in query frequency, while maintaining bounded computational overhead through single-threaded retrieval and early termination of outdated queries.

*Table 16.* Streaming Tool Queries generated by Qwen-OMNI after *Model-Triggered Stream RAG* post-training

| ASR Transcript of Question $X^{\mathbf{asr}}$ | Partial ASR Transcript $X_b^{\mathbf{asr}}$ | Web Query $\hat{Q}_b^{\mathbf{web}}$ | KG Query $\hat{Q}_b^{\mathbf{KG}}$ |
|---|---|---|---|
| Who founded Rare Beauty in 2019? | Who founded | Who founded what | {"domain": "other", "main_entity": "Who"} |
| | Who founded rare | Red Bull founder | {"domain": "other", "main_entity": "Red"} |
| | Who founded rare beauty | Who founded Rare Beauty | {"domain": "other", "main_entity": "Rare Beauty"} |
| | Who founded rare beauty in | NO_QUERY | NO_QUERY |
| | Who founded rare beauty in 2019? | NO_QUERY | NO_QUERY |
| How many jump shots did Darius Miles score with in the game on 2000-11-08? | how many | Number of queries on 03/27/2024 | {"domain": "other", "main_entity": "How many"} |
| | how many jump | Number of jump shots in basketball | {"domain": "other", "main_entity": "jump shot"} |
| | how many jump shots did | Derek Jeter jump shots count | {"domain": "sports", "sport_type": "other", "person": "Dairy", "datetime": "03/27/2024, 19:52:43 PT"} |
| | how many jump shots did Darius miles | Darius jump shots count | {"domain": "sports", "sport_type": "other", "person": "Darius", "datetime": "03/27/2024, 19:52:43 PT"} |
| | how many jump shots did Darius miles score | Darius Miles jump shots count | {"domain": "sports", "sport_type": "other", "person": "Darius Miles", "datetime": "03/27/2024, 19:52:43 PT"} |
| | how many jump shots did Darius miles score with in | Darius Miles jump shots scored | NO_QUERY |
| | how many jump shots did Darius miles score with in the game on | Darius Miles jump shots scored in game on 03/27/2024 | NO_QUERY |
| | how many jump shots did Darius miles score with in the game on November | Darius Miles jump shots scored in game on November 2024 | {"domain": "sports", "sport_type": "other", "person": "Darius Miles", "datetime": "November"} |
| | how many jump shots did Darius miles score with in the game on November 8 | Darius Miles jump shots scored on November 8 | {"domain": "sports", "sport_type": "other", "person": "Darius Miles", "datetime": "November 8"} |
| | how many jump shots did Darius miles score with in the game on November 8 | NO_QUERY | NO_QUERY |
| | how many jump shots did Darius miles score with in the game on November 8, 2000 | Darius Miles jump shots scored on November 8, 2000 | {"domain": "sports", "sport_type": "other", "person": "Darius Miles", "datetime": "November 8, 2000"} |

*Table 17.* Last-token Latency breakdown, showing median (P50) and 90th percentile (P90) timings, for the Qwen-OMNI in Open Book Setting on AudioCRAG-Synthetic (First Token Latency=5.9 sec in T. 1).

| Model | Token | P | Latency (sec) | | | |
| | | | Tool Latency | | Response Gen | Total |
| | | | Query Gen | Tool Results Gen | | |
| Qwen-OMNI | Last Token | P50 | 0.59 | 2.78 | 16.70 | 20.07 |
| | | P90 | 0.85 | 4.90 | 42.41 | 48.16 |

*Table 18.* Average number of tool calls for Model-Triggered Stream RAG on SLUE-SQA as a function of input duration.

| Audio Duration | Avg. Tool Calls | Examples (%) |
| --- | --- | --- |
| 0–3 s | 2.51 | 4.0 |
| 3–5 s | 3.74 | 36.4 |
| 5–7 s | 4.76 | 31.3 |
| 7–10 s | 5.70 | 20.2 |
| >10 s | 7.07 | 8.1 |
| Overall | 4.70 | 100.0 |

## A.17. Comparison with Concurrent Submission

AudioChat (Chen et al., 2026) is an E2E audio foundation model that performs multi-turn audio storytelling, editing, and understanding of complex multi-source acoustic scenes. Bagpiper (Tian et al., 2026) is a unified audio foundation model across speech, music, and sound that enables open-ended audio understanding and generation by learning a bidirectional mapping between audio and rich natural-language captions. While these approaches significantly expand the scope of audio tasks that can be solved end-to-end, they do not address the tool-integration challenges central to spoken dialogue systems. Specifically, they do not address when to invoke external tools during partial speech or how to minimize user-perceived latency, requiring them to either rely on potentially stale internal knowledge or defer retrieval until the user finishes speaking. Stream RAG is complementary to these efforts: it can be layered on top of such models to enable real-time, tool-augmented interaction, while introducing distinct modeling and evaluation questions centered on streaming tool integration and latency.

## A.18. Prompts used for Factual QA

### A.18.1. PROMPT IN CLOSED BOOK SETTING.

PROMPT = """ You are given an Audio Question and the time when it was asked in the Pacific Time Zone (PT), referred to as "Query Time". The query time is formatted as "mm/dd/yyyy, hh:mm:ss PT". Your task is to answer the question in as few words as possible.
Please follow these guidelines when formulating your answer:
1. If the question contains a false premise or assumption, answer "invalid question".
2. If you are uncertain or don't know the answer, respond with "I don't know".
### Question
{query}
### Query Time
{query_time}
### Answer
"""

### A.18.2. PROMPT IN OPEN BOOK / STREAM RAG SETTING.

PROMPT = """ You are given an Audio Question, References and the time when it was asked in the Pacific Time Zone (PT), referred to as "Query Time". The query time is formatted as "mm/dd/yyyy, hh:mm:ss PT". The references may or may not help answer the question. Your task is to answer the question in as few words as possible.
Please follow these guidelines when formulating your answer:
1. If the question contains a false premise or assumption, answer "invalid question".
2. If you are uncertain or don't know the answer, respond with "I don't know".
### Question

{query}
### Query Time
{query_time}
### References
# web
{web_results}
# knowledge graph
{kg_response}
### Answer
"""

A.18.3. KG QUERY EXTRACTION IN OPEN BOOK SETTING.

PROMPT = """" You are an agent that only outputs JSON. You are given a Query and Query Time. Do the following:

1) Determine the domain the query is about. The domain should be one of the following: "finance", "sports", "music", "movie", "encyclopedia". If none of the domains apply, use "other". Use "domain" as the key in the result json.

2) Extract structured information from the query. Include different keys into the result json depending on the domains, and put them DIRECTLY in the result json. Here are the rules:

For 'encyclopedia' and 'other' queries, these are possible keys:
- 'main_entity': extract the main entity of the query.

For 'finance' queries, these are possible keys:
- 'market_identifier': stock identifiers including individual company names, stock symbols.
- 'metric': financial metrics that the query is asking about. This must be one of the following: 'price', 'dividend', 'P/E ratio', 'EPS', 'marketCap', and 'other'.
- 'datetime': time frame that the query asks about. When datetime is not explicitly mentioned, use 'Query Time' as default.

For 'movie' queries, these are possible keys:
- 'movie_name': name of the movie
- 'movie_aspect': if the query is about a movie, which movie aspect the query asks. This must be one of the following: 'budget', 'genres', 'original_language', 'original_title', 'release_date', 'revenue', 'title', 'cast', 'crew', 'rating', 'length'.
- 'person': person name related to moves
- 'person_aspect': if the query is about a person, which person aspect the query asks. This must be one of the following: 'acted_movies', 'directed_movies', 'oscar_awards', 'birthday'.
- 'year': if the query is about movies released in a specific year, extract the year

For 'music' queries, these are possible keys:
- 'artist_name': name of the artist
- 'artist_aspect': if the query is about an artist, extract the aspect of the artist. This must be one of the following: 'member', 'birth place', 'birth date', 'lifespan', 'artist work', 'grammy award count', 'grammy award date'.
- 'song_name': name of the song
- 'song_aspect': if the query is about a song, extract the aspect of the song. This must be one of the following: 'author', 'grammy award count', 'release country', 'release date'.

For 'sports' queries, these are possible keys:
- 'sport_type': one of 'basketball', 'soccer', 'other'
- 'tournament': NBA, World Cup, Olympic.
- 'team': teams that users are interested in.
- 'datetime': time frame that the user is interested in. When datetime is not explicitly mentioned, use 'Query Time' as default.

Return the results in a FLAT json.

*NEVER include ANY EXPLANATION or NOTE in the output, ONLY OUTPUT JSON!!!*
"""

### A.18.4. KG QUERY EXTRACTION IN STREAM RAG SETTING.

PROMPT = """" You are an agent that only outputs JSON. You are given an Audio Query, Previously generated JSON result ('Previous Result') and Query Time. Do the following:

1) Determine the domain the query is about. The domain should be one of the following: ̈finance ̧ ̈sports ̧ ̈music ̧ ̈movie ̧ ̈encyclopedia ̈. If none of the domains apply, use ̈other ̈. Use ̈domain ̈as the key in the result json.

2) Extract structured information from the query. Include different keys into the result json depending on the domains, and put them DIRECTLY in the result json. Here are the rules:

For 'encyclopedia' and 'other' queries, these are possible keys: - 'main_entity': extract the main entity of the query.

For 'finance' queries, these are possible keys: - 'market_identifier': stock identifiers including individual company names, stock symbols. - 'metric': financial metrics that the query is asking about. This must be one of the following: 'price', 'dividend', 'P/E ratio', 'EPS', 'marketCap', and 'other'. - 'datetime': time frame that the query asks about. When datetime is not explicitly mentioned, use 'Query Time' as default.

For 'movie' queries, these are possible keys: - 'movie_name': name of the movie - 'movie_aspect': if the query is about a movie, which movie aspect the query asks. This must be one of the following: 'budget', 'genres', 'original_language', 'original_title', 'release_date', 'revenue', 'title', 'cast', 'crew', 'rating', 'length'. - 'person': person name related to moves - 'person_aspect': if the query is about a person, which person aspect the query asks. This must be one of the following: 'acted_movies', 'directed_movies', 'oscar_awards', 'birthday'. - 'year': if the query is about movies released in a specific year, extract the year

For 'music' queries, these are possible keys: - 'artist_name': name of the artist - 'artist_aspect': if the query is about an artist, extract the aspect of the artist. This must be one of the following: 'member', 'birth place', 'birth date', 'lifespan', 'artist work', 'grammy award count', 'grammy award date'. - 'song_name': name of the song - 'song_aspect': if the query is about a song, extract the aspect of the song. This must be one of the following: 'author', 'grammy award count', 'release country', 'release date'.

For 'sports' queries, these are possible keys: - 'sport_type': one of 'basketball ̧ ̈ 'soccer ̧ ̈ 'other ̈ - 'tournament': NBA, World Cup, Olympic. - 'team': teams that users are interested in. - 'datetime': time frame that the user is interested in. When datetime is not explicitly mentioned, use 'Query Time' as default. Return the results in a FLAT json.
*NEVER include ANY EXPLANATION or NOTE in the output, ONLY OUTPUT JSON!!!*

3) Compare your newly generated result to the 'Previous Result'. **If your new result would be exactly the same as the 'Previous Result', output only NO_QUERY.** Return the results in a FLAT json.

Previous Result:
{prev_kg_query}
"""

### A.18.5. WEB QUERY EXTRACTION IN OPEN BOOK SETTING.

PROMPT = """" You are given an Audio Query and Query Time. Your task is to generate a web query that can be used to retrieve relevant web pages. Rewrite the following query into a short and succinct form, focusing on the main topic or domain (e.g. finance, sports, music, movie, encyclopedia), key entities mentioned (e.g. people, organizations, locations), and specific aspects of those entities (e.g. performance metrics, relationships, events). Ensure the rewritten query is clear, concise, and easy to understand. Note that simply outputting the original query is not acceptable. You must rephrase the query to make it more concise and focused on the key information that will help retrieve relevant web pages.

For 'finance' queries, focus on: - Company names or stock symbols - Financial metrics (e.g. price, dividend, P/E ratio, EPS, marketCap) - Specific timeframes or events; if no timeframe is specified, use the Query Time as default

For 'sports' queries, focus on: - Sports Type (eg. basketball, soccer) - Teams, players - Statistics or performance metrics (e.g. scores, wins, losses) - Specific events or tournaments (eg. NBA, World Cup, Olympic) - Time frame that the user is interested in; if no timeframe is specified, use the Query Time as default

For 'music' queries, focus on: - Artist names or song titles - Specific aspects of artist (eg. band name, birth place, birth date, lifespan, artist work, grammy award count, grammy award date) - Specific aspects of song (eg. author, grammy award count, release country, release date) - Music genres or categories - Specific awards or recognition (e.g. Grammy Awards, Billboard)

For 'movie' queries, focus on: - Movie titles or celebrity names - Movie genres or other categories like budget, language, release_date, revenue, cast, crew, rating, length - Specific aspects of celebrity like acted_movies, directed_movies, oscar_awards, birthday - Specific awards or recognition (e.g. Oscars) For 'other' queries, focus on: - Main entity or topic - Specific aspects or attributes of the entity

When rewriting the query, ensure that it captures all important information from the original question that could impact the retrieval results. Do not omit any crucial details, such as specific dates, locations, or relationships between entities. Also, do not invent any new details on your own. If necessary, use the Query Time to provide context for the query. The goal is to create a concise and accurate query that effectively conveys the user's intent and retrieves relevant information. *NEVER include ANY EXPLANATION or NOTE in the output, ONLY OUTPUT THE REWRITTEN QUERY!!!* """

A.18.6. WEB QUERY EXTRACTION IN STREAM RAG SETTING.

PROMPT = """"You are given an Audio Query, previously generated Web query ('Previous Result') and Query Time.

Your task is to generate a web query that can be used to retrieve relevant web pages. Rewrite the following query into a short and succinct form, focusing on the main topic or domain (e.g. finance, sports, music, movie, encyclopedia), key entities mentioned (e.g. people, organizations, locations), and specific aspects of those entities (e.g. performance metrics, relationships, events). Ensure the rewritten query is clear, concise, and easy to understand.

Note that simply outputting the original query is not acceptable. You must rephrase the query to make it more concise and focused on the key information that will help retrieve relevant web pages.

For 'finance' queries, focus on: - Company names or stock symbols - Financial metrics (e.g. price, dividend, P/E ratio, EPS, marketCap) - Specific timeframes or events; if no timeframe is specified, use the Query Time as default

For 'sports' queries, focus on: - Sports Type (eg. basketball, soccer) - Teams, players - Statistics or performance metrics (e.g. scores, wins, losses) - Specific events or tournaments (eg. NBA, World Cup, Olympic) - Time frame that the user is interested in; if no timeframe is specified, use the Query Time as default

For 'music' queries, focus on: - Artist names or song titles - Specific aspects of artist (eg. band name, birth place, birth date, lifespan, artist work, grammy award count, grammy award date) - Specific aspects of song (eg. author, grammy award count, release country, release date) - Music genres or categories - Specific awards or recognition (e.g. Grammy Awards, Billboard)

For 'movie' queries, focus on: - Movie titles or celebrity names - Movie genres or other categories like budget, language, release_date, revenue, cast, crew, rating, length - Specific aspects of celebrity like acted_movies, directed_movies, oscar_awards, birthday - Specific awards or recognition (e.g. Oscars)

For 'other' queries, focus on: - Main entity or topic - Specific aspects or attributes of the entity

When rewriting the query, ensure that it captures all important information from the original question that could impact the retrieval results. Do not omit any crucial details, such as specific dates, locations, or relationships between entities. Also, do not invent any new details on your own. If necessary, use the Query Time to provide context for the query. The goal is to create a concise and accurate query that effectively conveys the user's intent and retrieves relevant information. Now, compare the new web query to the previously generated web query ('Previous Result').

If the new query is similar enough to the previous web query (i.e., it effectively conveys the same user intent and would retrieve similar relevant information), output only *NO_QUERY*.

Previous Result:
{prev_web_query}
"""

## A.19. LLM as judge prompt

PROMPT=""""

Assume you are a human expert in grading predictions given by a model. You are given a question and a model prediction. Judge if the prediction matches the ground truth answer by following these steps: 1: Take it as granted that the Ground Truth is always correct. 2: If the Prediction indicates it is not sure about the answer, "score" should be "0"; otherwise, go the next step. 3: If the Prediction exactly matches the Ground Truth, "score" is 1. 4: If the Prediction does not exactly match the Ground Truth, go through the following steps and likely give a score as 0. 5: If the Ground Truth is a number, "score" is 1 if and only if the Prediction gives a number that almost exactly matches the ground truth. 6: If the Prediction is self-contradictory, "score" must be 0. 7: If the prediction is not answering the question, "score" must be 0. 8: If the prediction is a concise and correct summary of the ground truth, "score" is 1. 9: If ground truth contains a set of items, prediction must contain exactly same items for the score to be 1. 10: Otherwise, "score" is 0.

Output a JSON blob with an "explanation" field explaining your answer as short as possible and an "score" field with value 1 or 0.

You should make the judgment based on provided examples. Examples: Question: "which company has higher eps, btu or cma?" Ground Truth: "cma" Prediction: "it is not possible to determine which company has a higher eps." Output: "score": 0, "explanation": "The prediction is not sure about the answer."

Question: "who is taller, a or b?" Ground Truth: "a" Prediction: "The answer is a. a is of height 1.75 m and b is of height 1.82 m. So b is taller." Output: "score": 0, "explanation": "The prediction is self-contradictory."

Question: "who among patricia place and lana clarkson is younger?" Ground Truth: "lana clarkson" Prediction: "lana clarkson (born april 5, 1962) is younger than patricia clarkson (born december 29, 1959)." Output: "score": 1, "explanation": "The prediction gives the correct answer."

Question: "did lakers win more games than warriors in 2024-4?" Ground Truth: "yes" Prediction: "yes, lakers won more games than warriors in 2023-2024 season." Output: "score": 0, "explanation": "The prediction is not answering the question."

Question: "what is the height of the building where citigroup is headquartered?" Ground Truth: "151 m" Prediction: "the height is 915 feet (279 m)." Output: "score": 0, "explanation": "The prediction, 151 m, does not match the ground truth, 279 m."

Question: "who were the last 3 prime ministers of uk?" Ground Truth: "rishi sunak, liz truss, and boris johnson" Prediction: "1. liz truss, 2. boris johnson, 3.theresa may" Output: "score": 0, "explanation": "The prediction does not contain item, rishi sunak, that is in the grount truth."

Question: "who authored the taming of the shrew?" Ground Truth: "william shakespeare" Prediction: "william shakespeare" Output: "score": 1, "explanation": "The prediction exactly matches the ground truth."

Question: "who authored The Taming of the Shrew?" Ground Truth: "william shakespeare" Prediction: "w shakespeare" Output: "score": 1, "explanation": "The prediction matches the ground truth as w is the abbreviation of william."

Question: "what is the state bird of california?" Ground Truth: "california quail" Prediction: "california valley quail" Output: "score": 1, "explanation": "The prediction matches the ground truth as california quail is also known as california valley quail."

Question: "what country is maharashtra metro rail corporation limited located in?" Ground truth: "india" Prediction: "indianapolis" Output: "score": 0, "explanation": "The prediction doesn't match the ground truth answers."

Question: "what was wolves's record in their last game in eng-premier league?" Ground Truth: "w" Prediction: "wolves's record in their last game in the eng-premier League was a 2-1 win over fulham." Output: "score": 1, "explanation": "The prediction correctly answers the result is a win."

Question: "how many seconds is 3 minutes 15 seconds?" Ground Truth: "195 seconds" Prediction: "3 minutes 15 seconds is 195 seconds." Output: "score": 1, "explanation": "The prediction matches ground truth as it correctly states the number of seconds asked in the question."

Question: "on which days did am distribute dividends in the last year?" Ground Truth: "2023-01-24, 2023-04-25, 2023-07-25, 2023-10-24" Prediction: "the company distributed dividends on october 24, 2023 and january 24, 2023." Output: "score": 0, "explanation": "The prediction partially matches the ground truth, but it is incomplete."

Question: "on which days did xxx distribute dividends in the last year?" Ground Truth: "2023-01-13, 2023-03-25, 2023-11-21" Prediction: "xxx distributed dividends on 1. 2023-01-13, 2. 2023-03-25, 3. 2023-10-21." Output: "score": 0, "explanation": "The prediction contains an item, 2023-10-21, which is not in ground truth."

Question: "who are current band members of maroon 5?" Ground Truth: "adam Levine, jesse carmichael, james valentine, matt flynn, pj morton, sam farrar" Prediction: "they are jesse carmichael, matt flynn, adam levine, pj morton, sam farrar, and james valentine." Output: "score": 1, "explanation": "The prediction exactly matches the ground truth."

Question: "which movies comprise the matrix franchise?" Ground Truth: "the matrix, the matrix reloaded, the matrix revolutions, the matrix resurrections" Prediction: "the matrix, the matrix reloaded, the matrix revolutions, the animatrix, and the matrix resurrections." Output: "score": 0, "explanation": "The prediction covers more items than what are given by the ground truth."

Question: "how deep is the deepest lake of new york?" Ground Truth: "618 ft" Prediction: "the deepest lake in new york is seneca lake, with a depth of 618.23 feet." Output: "score": 1, "explanation": "The prediction exactly matches the number in ground truth after rounding."

Question: "what is the closing price of meta yesterday?" Ground Truth: "$310.17" Prediction: "310.2" Output: "score": 1, "explanation": "The prediction exactly matches the number in ground truth after rounding."

Question: "what is the current market cap of appl?" Ground Truth: "2.81 trillion" Prediction: "2.667 trillion" Output: "score": 0, "explanation": "The prediction does not match the number in ground truth."

Question: "what is the current pe ratio of appl?" Ground Truth: "28.3" Prediction: "the current pe ratio of apple is 26.66" Output: "score": 0, "explanation": "The prediction does not match the number in ground truth."

Question: "how much is tesla's stock price down from its all-time high?" Ground Truth: "$221.83" Prediction: "209.52" Output: "score": 0, "explanation": "The prediction does not match the number in ground truth."

Question: "what is the length of amazon river?" Ground Truth: "over 4000 miles" Prediction: "the length of amazon river is 4,000 miles" Output: "score": 0, "explanation": "The prediction does not say Amazon River is longer than 4000 miles."

Question: "how many copies x were sold?" Ground Truth: "2 million." Prediction: "it is over 2 million" Output: "score": 0, "explanation": "The prediction does not match the ground truth."

Question: "what is the population of country x?" Ground Truth: "3,576,873" Prediction: "the population of country x is 3.3 million." Output: "score": 0, "explanation": "The prediction, 3.3 M, does not match the number, 3.6 M, in ground truth."

Question: "what is the current market value of stock x?" Ground Truth: "$2,237,578,268" Prediction: "$2.16 billion." Output: "score": 0, "explanation": "The prediction, 2.16 B, does not match the number, 2.2 B, in ground truth."

Question: "how old is michael jordan?" Ground Truth: "61" Prediction: "mj was born on February 17, 1963." Output: "score": 0, "explanation": "The prediction does not explicitly say the age."

Question: "how long did amy teach as a teacher?" Ground Truth: "24 years and 255 days" Prediction: "25 years." Output: "score": 0, "explanation": "The prediction does not exactly match the ground truth."

Question: "at what age did usher perform in the half time show of superbowl?" Ground Truth: "45 years old" Prediction: "usher has not performed at a super bowl halftime show." Output: "score": 0, "explanation": "The prediction does not match the ground truth."

Question: "what year did olivia rodrigo win the grammys best new artist award?" Ground Truth: "2022" Prediction: "sure! here's an answer in three words or fewer: 2022 grammy awards." Output: "score": 1, "explanation": "The prediction answers the correct number of the year."

Question: "how many times have zendaya and tom holland starred in a film together?" Ground Truth: "3" Prediction: "they have starred in 3 films together: spider-man: homecoming, spider-man: far from home, and spider-man: no way home." Output: "score": 1, "explanation": "The prediction gives the right number and it is reasonable."

Question: "who is younger, timothee chalamet or tom holland?" Ground Truth: "tom holland" Prediction: "timothée chalamet is younger than tom holland." Output: "score": 0, "explanation": "The prediction does not match the ground truth."

Question: "who had more number one hits on the us billboard, a or b?" Ground Truth: "a had more number one hits on the us billboard than b, with 20 number one hits compared to b's 15." Prediction: "a" Output: "score": 1, "explanation": "The prediction is a concise and correct summary of the ground truth."

Question: "what is xxx's birthdate?" Ground Truth: "1996-01-01." Prediction: "02/01/1996" Output: "score": 0, "explanation": "The prediction does not match the ground truth."

Question: "what was the worldwide box office haul for movie x?" Ground Truth: "101756123." Prediction: "102 million" Output: "score": 1, "explanation": "The prediction exactly matches the number in ground truth after rounding."

Question: "how much has spotify's user base increased by since 2020 in na?" Ground Truth: "spotify's user base increased by 34 million since 2020." Prediction: "spotify's north american user base increased from 36 million in 2020 to 85 million by 2021" Output: "score": 0, "explanation": "The prediction is not answering the question as it only gives the increase from 2020 to 2021."
"""

*Table 19.* Post-training Parameters of OpusLM

| Parameter | Value |
| --- | --- |
| train_micro_batch_size_per_gpu | 1 |
| gradient_accumulation_steps | 2 |
| epochs | 2 |
| gradient_clipping | 1.0 |
| bf16 enabled | true |
| optimizer type | Adam |
| optimizer lr | 0.00001 |
| optimizer betas | [0.9, 0.95] |
| optimizer eps | 1e-8 |
| optimizer weight_decay | 3e-7 |
| optimizer adam_w_mode | true |
| scheduler type | WarmupDecayLR |
| scheduler warmup_type | linear |
| scheduler total_num_steps | 21534 |
| scheduler warmup_num_steps | 1077 |
| scheduler warmup_min_lr | 0 |
| scheduler warmup_max_lr | 0.00001 |

*Table 20.* Post-training Parameters of Qwen-OMNI Thinker

| Parameter | Value |
| --- | --- |
| bf16 | True |
| gradient_accumulation_steps | 4 |
| epochs | 1 |
| gradient_clipping | 1.0 |
| learning_rate | 7e-6 |
| lr_scheduler_type | cosine |
| warmup_ratio | 0.05 |
| per_device_train_batch_size | 1 |
| weight_decay | 0.01 |

*Table 21.* Post-training Parameters of Qwen-OMNI Talker

| Parameter | Value |
|---|---|
| bf16 | True |
| gradient_accumulation_steps | 4 |
| gradient_clipping | 1.0 |
| epochs | 2 |
| learning_rate | 5e-5 |
| per_device_train_batch_size | 1 |
| lr_scheduler_type | linear |
| warmup_ratio | 0.0 |
| weight_decay | 0.01 |

