# OpenReview forum: "Stream RAG: Instant and Accurate Spoken Dialogue Systems with Streaming Tool Usage"
_ICML.cc/2026/Conference — ICML 2026 regular_

### Official Review · Reviewer_jpRJ · 2026-02-24

**Soundness:** 3
**Presentation:** 3
**Significance:** 3
**Originality:** 3
**Overall Recommendation:** 5
**Confidence:** 3

**Summary:**

This paper proposes Stream RAG, a streaming retrieval-augmented generation framework for end-to-end speech-to-speech SDS that addresses the inherent trade-off between factual accuracy and response latency. While integrating external tools such as web search can improve grounding and reduce hallucinations in dialogue systems, conventional RAG pipelines introduce additional delays because retrieval is performed only after the user finishes speaking. Stream RAG resolves this issue by predicting retrieval queries from partial speech inputs during ongoing user utterances and executing tool calls in parallel with speech processing. The retrieved textual information is then incorporated into real-time spoken responses through a post-training pipeline that enhances tool-usage capabilities of pretrained speech-in speech-out dialogue models without requiring full end-to-end retraining. Experimental results on the proposed AudioCRAG benchmark demonstrate that Stream RAG significantly improves question-answering accuracy while reducing end-to-end system latency compared to both cascade and non-streaming RAG-based SDS baselines.

**Compliance With Llm Reviewing Policy:**

Affirmed.

**Final Justification:**

The rebuttal was sufficiently reasonable, so the score was increased.

**Key Questions For Authors:**

See Weaknesses.

**Limitations:**

yes

**Strengths And Weaknesses:**

Strengths

1. A key strength of this work lies in its principled approach to overcoming the latency–accuracy trade-off in retrieval-augmented spoken dialogue systems by enabling proactive and streaming tool usage before utterance completion. Unlike prior approaches that rely on sequential pipelines or require costly end-to-end retraining, Stream RAG improves factual grounding and responsiveness through lightweight post-training that teaches pretrained multimodal SDS models when and how to invoke external tools during real-time interaction. The introduction of the AudioCRAG benchmark further strengthens the study by enabling realistic evaluation in speech-based QA settings. Moreover, the proposed framework is modular and model-agnostic, making it broadly applicable to existing speech-enabled large language models and practical for deployment in real-time AI assistants where both low latency and high accuracy are critical.

Weaknesses

1. One limitation of this work is that the experimental comparisons may not be entirely fair, as the baselines reported in the main results are used without any additional training, whereas Stream RAG benefits from post-training that explicitly enhances its tool-usage capabilities. This makes it difficult to disentangle whether the observed performance gains stem from the proposed streaming retrieval paradigm itself or simply from the added training procedure, suggesting that comparisons with other trained or fine-tuned RAG-based spoken dialogue systems would provide a more rigorous evaluation.
2. Additionally, the core assumption of predicting tool usage during an ongoing user utterance introduces a potential robustness issue: if the user’s intent becomes clear only toward the end of the utterance, early query prediction may trigger retrieval from an inappropriate tool or knowledge source, potentially leading to irrelevant or suboptimal responses. This highlights an inherent limitation of proactive tool invocation in streaming settings, where premature decisions may negatively impact downstream response quality.

---

> ### Author Rebuttal · Authors · 2026-03-31
>
> We thank the reviewer for their positive recommendation and acknowledging our work as a “principled approach to overcoming the latency–accuracy trade-off”. We address the remaining concerns below.
>
> > The baselines are used without additional training, whereas Stream RAG benefits from post-training. The accuracy gains may stem from post-training rather than the streaming architecture.
>
> A: We thank the reviewer for this important point and agree that post-training is the primary driver of accuracy gains. **Our goal is to isolate whether the streaming architecture preserves this accuracy while reducing latency, which we explicitly evaluate in our ablation study (Table 5).**
> Specifically, the “Tool Integration (Sec. 4.2)” row corresponds to a post-trained sequential RAG baseline (i.e., post-training without streaming). For Qwen-OMNI, this achieves 34.9% accuracy, compared to 34.2% for Stream RAG, showing that the streaming formulation maintains comparable accuracy to its sequential counterpart.
> The key takeaway is that Stream RAG’s contribution is not improved accuracy, but achieving similar accuracy at substantially lower latency (up to 57% reduction, Sec. 5.4). In contrast, the gains over the zero-shot Open Book baseline in Table 1 are indeed largely attributable to post-training.
> We agree that this distinction should be made more explicit. In the revision, we will prominently include the post-trained sequential RAG baseline in Table 1 to clearly separate the effects of post-training and streaming.
>
> > If the user’s intent becomes clear only toward the end of the utterance, early query prediction may trigger retrieval from an inappropriate tool or knowledge source, potentially leading to irrelevant or suboptimal responses.
>
> A: This is a key challenge in streaming retrieval, and **Stream RAG is explicitly designed to mitigate it through learned triggering, recovery from incorrect early queries, and a bounded fallback to standard RAG behavior**.
>
> (1) Learned triggering. In Model-Triggered Stream RAG, the model learns when not to issue a query. Through similarity-based labeling (Eq. 4), it is trained to generate a new query only when additional useful information has emerged, and to emit a “NO QUERY” token otherwise.
>
> (2) Recovery from premature queries. Our negative sampling strategy (Eq. 5) explicitly trains the model to recover from incorrect early queries by overriding them as more speech becomes available.
>
> (3) Empirical robustness. Consistent with this design, early tool invocation does not correlate with increased error rates (Sec. 5.2 and A.6, Table 9). Manual inspection (see Table 11) further shows that most remaining errors stem from query formulation or answer selection rather than premature tool invocation, and qualitative examples in Table 15 illustrate that the model can refine or correct early queries as the utterance unfolds.
>
> (4) Graceful revision. In many cases, if an early retrieval is unhelpful, the model can still revise the query once more information becomes available, bringing behavior closer to standard sequential RAG rather than locking into the initial mistake. To further quantify robustness, we conducted an additional perturbation experiment on SLUE-SQA (S. A.3), where the previous query \hat{Q}^{\text{prev}}_b is replaced with a negative sampled query. Even under this challenging setting, Stream RAG achieves 65.4% accuracy, showing that the model can still recover the correct final retrieval outcome despite intermediate query errors.
>
> We will clarify this robustness analysis and include additional discussion of qualitative examples in the revised paper.

---

> > ### Author Rebuttal · Reviewer_jpRJ · 2026-04-03
> >
> > Thank you for your response. The rebuttal was sufficiently reasonable, so the score was increased.

---

> > > ### Author Response · Authors · 2026-04-03
> > >
> > > We sincerely thank Reviewer jpRJ for the positive feedback and for recognizing our efforts in addressing the concerns. We are glad that our clarifications were helpful. Thank you for your time and constructive support.

---

### Official Review · Reviewer_acrd · 2026-03-11

**Soundness:** 3
**Presentation:** 4
**Significance:** 4
**Originality:** 3
**Overall Recommendation:** 5
**Confidence:** 3

**Summary:**

This paper introduces Stream RAG, a novel framework designed to enhance e2e spoken dialogue systems with real-time tool usage capabilities. While e2e systems provide more natural interactions than traditional cascaded pipelines, they often suffer from hallucinations. Integrating RAG typically introduces significant latency, disrupting the conversational flow. Stream RAG addresses this by parallelizing the retrieval process during the user's speech, triggering search queries as the audio stream is processed. The authors propose two strategies: Fixed-Interval and Model-Triggered triggering. Experimental results on a new benchmark, AudioCRAG, demonstrate that Stream RAG improves factual accuracy while reducing first-token latency by up to 57% compared to sequential RAG approaches.

**Compliance With Llm Reviewing Policy:**

Affirmed.

**Final Justification:**

The response effectively addressed my primary concerns regarding the efficiency and operational logic of the Model-Triggered Stream RAG, particularly the clarification that only a single tool call thread remains active at any given time to manage computational overhead. While the average of 4.7 tool calls per utterance is significantly higher than the traditional baseline, I agree that this is a reasonable and well-managed trade-off for the substantial latency gains achieved in real-time voice interactions. I believe this work makes a meaningful and practical contribution to the field, and I have therefore increased my score to reflect my updated assessment.

**Key Questions For Authors:**

My comments are detailed in the Weaknesses part. I have no additional questions.

**Limitations:**

yes

**Strengths And Weaknesses:**

# Strength
1. The paper addresses a critical pain point in spoken dialogue systems: the trade-off between factual accuracy and low latency. The motivation to hide retrieval latency within the user’s speaking time is highly intuitive and addresses a major barrier to deploying RAG in real-time e2e voice AI.
2. The transition from sequential listen-then-query to parallel listen-and-query is a good architectural improvement.
3. The empirical results are impressive, showing a substantial reduction in latency, up to 57%, without compromising accuracy. The ability to achieve negative latency is a breakthrough for user experience.

# Weakness
My primary concern is the frequency of tool calls. For instance, a 30-second user utterance might trigger multiple RAG processes. When comparing the traditional RAG baseline (which waits for the full request before retrieving) with Stream RAG, is there a difference in the total number of tool calls? Intuitively, Stream RAG likely requires as many or more tool calls than the traditional approach. While traditional RAG can design precise queries or sub-questions based on the complete request, Model-Triggered Stream RAG acts more like a greedy strategy. It triggers a search as soon as a need is detected, but in real-world scenarios, the user might provide additional details in the following seconds, necessitating a new or revised search. I suggest the authors provide a comparison of the tool call frequency between Stream RAG and the traditional baseline. This would offer a valuable efficiency perspective beyond just latency.

While the Stream RAG architecture theoretically supports multiple queries, it is unclear whether the current experiments sufficiently test its performance in such cases. For long requests requiring multiple searches, is it possible that Stream RAG misses critical retrieval points? Does the AudioCRAG dataset include scenarios that require multiple tool calls for a single user request? If the authors could provide more statistical details about the dataset regarding this, it would greatly help readers understand the upper performance limit of Stream RAG in complex, real-world interactions.

---

> ### Author Rebuttal · Authors · 2026-03-31
>
> We thank the reviewer for their positive recommendation and acknowledging our work “addresses a major barrier to deploying RAG in real-time e2e voice AI”. We answer the remaining concerns below.
>
> > Is there a difference in the total number of tool calls between Stream RAG and traditional RAG? Stream RAG likely requires more tool calls, Model-Triggered Stream RAG acts more like a greedy strategy that may need revised searches as users provide more detail.
>
> A: **Yes, Stream RAG can issue more tool calls than traditional RAG, but this is an intentional trade-off for latency reduction, and in the Model-Triggered setting it remains computationally controlled.** For Fixed-Interval Stream RAG, more tool calls are expected since queries are triggered at every fixed interval. However, these calls are parallelized rather than sequential, so they do not add user-perceived latency in the same way as repeated sequential retrieval. Moreover, the Reflector ensures that only the earliest useful query result is ultimately used for response generation.
>
> For Model-Triggered Stream RAG, the model is more selective and can decide whether to issue a new tool query at each fixed interval. It learns to trigger a new query only when additional information becomes available, rather than at every interval. While this can still result in multiple query updates for longer utterances, the system maintains only a single active tool thread at any time: as soon as a new query is generated, any ongoing tool call for the previous query is immediately terminated. Thus, the method does not accumulate multiple full retrieval pipelines in parallel, and behaves more like query refinement over time than unconstrained greedy search.
>
> Empirically, on SLUE-SQA, Model-Triggered Stream RAG issues an average of 4.7 tool calls per utterance, compared to 1.0 for traditional RAG. While this is a higher call count, the practical overhead remains modest, since only one tool call thread is active at a time. As expected, this number increases with utterance duration:
>
> | Audio Duration | Avg. Tool Calls | Examples (%) |
> |---|---:|---:|
> | 0–3 s | 2.51 | 4.0 |
> | 3–5 s | 3.74 | 36.4 |
> | 5–7 s | 4.76 | 31.3 |
> | 7–10 s | 5.70 | 20.2 |
> | >10 s | 7.07 | 8.1 |
>
> We will add this detailed comparison of tool call frequency in the revised paper.
>
> > For long requests requiring multiple searches, does Stream RAG risk missing critical retrieval points? Does AudioCRAG include multi-tool-call scenarios?
>
> A: **Stream RAG’s design mitigates this risk by enabling iterative query refinement as more information becomes available.**
> For long or evolving queries, the Model-Triggered approach can issue multiple tool calls over time, updating queries as new information is revealed in the user’s speech. This allows early coarse queries followed by more precise ones as the utterance unfolds, whereas traditional RAG typically commits after the full utterance.
>
> Regarding evaluation, we agree that AudioCRAG primarily focuses on single-turn factual QA, where a single retrieval is often sufficient. However, we do evaluate multi-turn and context-dependent scenarios in Appendix A.4 using the CORAL multi-turn conversational benchmark, where user intent depends on prior turns. As shown, Stream RAG maintains comparable response quality while achieving 1.83s latency reduction on subsequent turns, demonstrating that it can effectively handle multi-turn, context-dependent retrieval without degrading performance .
>
> We agree that **explicitly extending AudioCRAG with multi-query / multi-hop retrieval questions is an important direction, and we will clarify this limitation and include such extensions in future work**.

---

> > ### Author Rebuttal · Reviewer_acrd · 2026-04-03
> >
> > The authors have clarified the latency-efficiency trade-off and provided valuable empirical data. My concerns are addressed, and I am happy to raise my score.

---

> > > ### Author Response · Authors · 2026-04-04
> > >
> > > We sincerely thank Reviewer acrd for the positive feedback and for recognizing our efforts in addressing the concerns. We are glad that our clarifications were helpful. Thank you for your time and constructive support.

---

### Official Review · Reviewer_Wzdi · 2026-03-11

**Soundness:** 3
**Presentation:** 3
**Significance:** 3
**Originality:** 2
**Overall Recommendation:** 4
**Confidence:** 4

**Summary:**

The paper proposes Streaming Retrieval-Augmented Generation (Stream RAG) for end-to-end speech-in, speech-out dialogue systems to reduce hallucinations while maintaining low latency. The key idea is to predict retrieval queries during the user’s speech, allowing the system to start tool calls (e.g., web search) before the user finishes speaking, thereby hiding retrieval latency. The authors train the model via a post-training pipeline that teaches when to trigger tools and how to generate spoken responses from retrieved text. Experiments on the newly created AudioCRAG benchmark and SLUESQA show that Stream RAG improves QA accuracy by over 20% and reduces latency by up to 57%, outperforming cascaded ASR–LLM–TTS systems.

**Compliance With Llm Reviewing Policy:**

Affirmed.

**Final Justification:**

The primary concern with this paper lies in its novelty and depth of insight. Overall, the work primarily focuses on engineering improvements across multiple components to enable streaming RAG. While the reviewer acknowledges the importance of such engineering efforts, the paper lacks sufficiently novel or insightful technical contributions.

The authors are also encouraged to include a discussion of the scope, particularly clarifying how their approach relates to streaming models such as Moshi, NTPP, and LLaMA-Omni, which do not inherently possess the capabilities targeted in this work.

That said, the addition of human evaluation (which should be included in the main paper) and the generally solid experimental validation strengthen the submission. As a result, the reviewer is willing to increase the score to a weak accept.

**Key Questions For Authors:**

(1) How does the model determine when sufficient partial speech has been observed to generate a reliable retrieval query? Is there an explicit confidence threshold or triggering policy that decides when to issue a tool call, particularly in the fixed-interval setting?

(2) The proposed parallel RAG framework appears to rely on an ASR–TTS translation step to process tool retrieval. How sensitive is the system to inaccuracies in these auxiliary cascading components? In particular, what impact do errors from this “small cascading pipeline” have on the overall retrieval quality and final response generation?

**Limitations:**

Authors discuss some important limitations. This reviewer feels there are some other limitations. First, the trigger model training is non-trivial, which is very similar to VAD module. Second, although the proposed method is applied to streaming model, the parallel tool call is a lightweight cascading pipeline, in which each stage accuracy would affect the final retrieval result.

**Strengths And Weaknesses:**

Strengths

(1) The paper is well written and well organized. The core idea is clearly presented, making it relatively easy for readers to understand the main motivation and approach.

(2) Improving the reliability of speech-based interactions is an important and practical problem. The reviewer appreciates the effort to address this challenge in real-world speech dialogue systems.

(3) The empirical results demonstrate a meaningful improvement in latency, suggesting that the proposed approach could provide practical benefits for real-time conversational systems.

Weaknesses

(1) Human evaluation is missing from the experimental results. Given that the task focuses on speech-based dialogue systems, human evaluation would be important for assessing response quality, naturalness, and overall user experience. In particular, latency is a nuanced metric in this context, as reductions in latency should be evaluated alongside the quality and usefulness of the generated responses, rather than in isolation.

(2) The novelty of the proposed method appears somewhat limited. The main contribution seems to lie more in system engineering and integration rather than in introducing a fundamentally new methodological framework. While the approach may be useful for practical deployment, the level of methodological novelty may not fully meet the typical expectations for ICML.

---

> ### Author Rebuttal · Authors · 2026-03-31
>
> > Human evaluation is missing. Latency reductions should be evaluated alongside quality and usefulness of generated responses.
>
> A: We agree that human evaluation is important for speech dialogue systems. **To directly address this concern, we conducted a human evaluation on 100 samples (200 ratings, 2 annotators/sample) comparing Stream RAG against a Cascade baseline on SLUE SQA dataset (see Table 2).** Annotators rated (i) Answer Correctness vs. Ground Truth (5 = perfectly correct, 1 = completely wrong) and (ii) Response Latency / silence before answer (5 = instant, 1 = very long delay).
>
> | Metric                        | Stream RAG | Cascade | Δ     |
> |------------------------------|-----------:|--------:|------:|
> | Correctness (mean)    | 3.75       | 2.71    | **+1.04** |
> |  Latency (mean)      | 4.27       | 3.15    | **+1.12** |
>
> We also collected overall pairwise preference, where Stream RAG was preferred in 74.5% of ratings vs 10.0% for the cascade baseline (15.5% no preference), with 77% inter-annotator agreement. These results indicate that Stream RAG improves both perceived response quality and latency, and that these gains translate into a clear overall user preference. We will include these results in the revision.
>
> > The novelty appears limited-the contribution seems to lie more in system engineering than in a fundamentally new methodological framework.
>
> A: We appreciate this concern and would like to clarify the novelty more precisely. The key contribution is not simply adding retrieval to a speech system, but **introducing a new problem formulation: streaming tool invocation under partial observations**. Prior speech RAG systems such as WavRAG assume the full utterance is available before retrieval, whereas Stream RAG enables tool queries while the user is still speaking. This requires several non-trivial methodological contributions:
> 1. a new training paradigm for deciding when to issue tool queries under incomplete input, addressed via similarity-based labeling (Eq. 4) and negative sampling (Eq. 5);
> 2. a structured inference-time framework (Trigger / Threads / Reflector) for proactive tool use during ongoing speech; and
> 3. AudioCRAG, to our knowledge the first spoken QA benchmark designed for tool-augmented speech dialogue.
>
> Importantly, Stream RAG’s primary architectural contribution is latency reduction while maintaining the accuracy gains from post-training. This is supported by Table 5, which shows that the proposed streaming formulation preserves the accuracy of post-trained sequential RAG while substantially reducing latency, and by Table 13, which shows that negative sampling is critical for maintaining performance under partial input. We will revise the paper to more clearly highlight the methodological novelty.
>
> > How does the model determine when sufficient partial speech has been observed? Is there an explicit confidence threshold or triggering policy in the fixed-interval setting?
>
> **In short, Fixed-Interval uses no learned trigger and instead relies on a post-hoc Reflector, whereas Model-Triggered learns an implicit policy for when to issue a new tool query.**
> For Fixed-Interval, tool queries are issued at predefined intervals; a Reflector then selects the earliest intermediate query whose retrieved evidence matches the final query (e.g., same top-5 web documents). For Model-Triggered, the model is trained to emit either “NO QUERY” or a new tool query. Similarity-based labeling (Eq. 4) identifies when a new query is unnecessary, while negative sampling (Eq. 5) teaches the model to recover from premature or incorrect early queries. Thus, no explicit confidence threshold is required.
>
> > The proposed framework appears to rely on an ASR–TTS translation step to process tool retrieval. How sensitive is the system to inaccuracies in these auxiliary cascading components?
>
> A: **Stream RAG is not a standard ASR→LLM→TTS cascade. The only intermediate text is the retrieval query; the final response is still generated end-to-end from speech and retrieved evidence, limiting error propagation.** Empirically, the main bottleneck is speech realization, not retrieval robustness: in Table 3, text-output performance remains nearly unchanged under Stream RAG (e.g., 39.8% vs. 39.6% for Qwen-OMNI), while the larger degradation appears only in spoken output (34.2%), consistent with the known text–speech modality gap (Table 8). Moreover, with text-in/text-out (i.e., effectively perfect ASR/TTS), Qwen-OMNI reaches only 38.9% accuracy, indicating that our method already operates close to the underlying LLM’s accuracy ceiling. Robustness is further improved by our negative sampling strategy, which explicitly trains the model to recover from incorrect intermediate queries as discussed in our response to reviewer jpRJ.

---

> > ### Author Rebuttal · Reviewer_Wzdi · 2026-04-04
> >
> > Thank you for your response. The reviewer provides the following comments:
> >
> > (1) Please include human evaluation results in the experimental section and provide sufficient description and analysis.
> >
> > (2) The reviewer remains concerned about the novelty and technical contribution. Although the authors claim that the streaming setting introduces a new paradigm for audio RAG and poses additional challenges, it is still unclear what fundamentally distinguishes this setting and why it is particularly challenging. The authors are encouraged to clearly summarize the novel challenges and the specific problems addressed in this work.
> >
> > Furthermore, from a rigorous standpoint, such a framework should be evaluated on streaming full-duplex models (e.g., Moshi, NTPP, and LLaMA-Omni). The reviewer would like to understand why these representative streaming audio models are not included in the evaluation.

---

> > > ### Author Response · Authors · 2026-04-04
> > >
> > > Thank you again for the helpful follow-up. The reviewer’s feedback has helped us identify a genuine presentation gap in the paper: while the method was designed around a new streaming decision problem, we did not state that contribution explicitly enough. We will revise this substantially in the main paper.
> > >
> > > **To directly answer the reviewer’s request to summarize the novel challenges and the specific problems addressed:** our contribution is not simply “RAG for audio”, but a **new setting: tool invocation under partial speech.** Unlike standard RAG (retrieve after endpoint), Stream RAG must decide whether / when / how to retrieve during an incomplete utterance. This introduces challenges that are both qualitatively new and empirically non-trivial:
> > >
> > > 1. **Decision-making under partial observability (when to act):** the model must learn when partial input contains enough information to issue a useful query vs. when to wait. This is formalized through our similarity-based labeling strategy (Eq. 4), which provides training signal from partial observations — a problem with no direct counterpart in endpoint-only text/audio RAG.
> > > 2. **Premature / incorrect queries → error propagation**, addressed via negative sampling (Eq. 5); without it, accuracy drops (−3.3% ablation in Table 13).
> > > 3. **Latency–accuracy tradeoff:** Stream RAG introduces a tradeoff that does not exist in endpoint-only retrieval, retrieving earlier can reduce delay but risks harming correctness. Our learned policy resolves this effectively: accuracy remains stable across latency-saving regimes, including ~52–58% EM even under aggressive early triggering (>3s saved, Table 9).
> > >
> > > These challenges are absent in endpoint-only retrieval and require explicit modeling of online decision-making under partial observability, which is the core methodological contribution.
> > >
> > > Regarding Moshi / NTPP / LLaMA-Omni: these systems **do not support external tool invocation**, i.e., they do not provide a mechanism to generate retrieval queries from partial speech or to condition response generation on retrieved external evidence. This is therefore not simply an omitted baseline, but a **mismatch with the capability under evaluation**. Since the central problem in this paper is streaming tool use, not just streaming generation, they are not valid baselines for this capability. Our comparisons instead isolate the most relevant axis: endpoint-only vs. streaming retrieval in speech-in/speech-out systems. We will clarify this scope distinction explicitly in the revision.
> > >
> > > On human evaluation, as the reviewer rightly noted, latency reductions should be evaluated alongside response quality. We have already conducted this evaluation and will move it into the main paper with full details. The results show consistent gains in both usefulness and responsiveness: **+1.04 correctness (3.75 vs. 2.71), +1.12 latency rating, and 74.5% overall preference** (vs. 10.0%) with 77% inter-annotator agreement. These results show that the latency gains are not merely system-level improvements, but translate into a clearly better user experience.
> > >
> > > Concretely, in the revision we will:
> > > (1) add a **short “Challenges of Streaming Tool Invocation” paragraph in §3.2**,
> > > (2) make the novelty / challenge framing explicit already in the Introduction, and
> > > (3) move the human evaluation from rebuttal into the section 5.6 with sufficient description and analysis.
> > >
> > > We hope these clarifications resolve the remaining concerns and respectfully ask the reviewer to reconsider the score.

---

### Official Review · Reviewer_HhuX · 2026-03-12

**Soundness:** 3
**Presentation:** 3
**Significance:** 3
**Originality:** 3
**Overall Recommendation:** 5
**Confidence:** 4

**Summary:**

This paper addresses the latency limitations that arise when incorporating Retrieval-Augmented Generation (RAG) techniques to enhance the conversational quality of end-to-end (E2E) spoken dialogue systems. Specifically, it proposes a method whereby the model autonomously determines the appropriate timing for tool calls, along with a post-training approach (model-triggered) to enable this capability and a benchmark called AudioCRAG designed for evaluation purposes. Through these contributions, the paper demonstrates significant improvements in dialogue performance while simultaneously reducing latency.

**Compliance With Llm Reviewing Policy:**

Affirmed.

**Final Justification:**

After the rebuttal period, I remain inclined toward a positive evaluation.

**Key Questions For Authors:**

I have noted two questions above.

**Limitations:**

yes

**Strengths And Weaknesses:**

**Strengths:**

I personally believe that the direction and focus of this paper are truly important and necessary. Ultimately, there are several reasons why end-to-end speech LLMs still compete with cascaded approaches, unlike VLMs in the vision domain, which have been trained to naturally accept image and video inputs, including performance, pluggability, and latency. However, I consider one of the major factors to be how these systems handle the components required for agentic capabilities, such as retrieval and tool calling. Given this context, the approaches proposed to reduce latency, one of the most significant bottlenecks, by checking intermediate values at midpoints or by enabling the model to autonomously determine the appropriate timing for tool calls appear to be a sufficiently valuable direction.

Furthermore, the authors convincingly demonstrate actual performance improvements and latency reduction through their experimental results. The manner in which the results are presented and the overall flow of the paper are natural and coherent, making this a well-written and commendable piece of research.

**Weaknesses & Questions:**

Rather than a weakness, I have a personal question: I am curious whether there are any plans to release the training data or models. It seems that, particularly in domains such as speech, publicly available RAG-related models and datasets remain remarkably scarce. Given this situation, I would like to ask whether there are any plans, though by no means an obligation, to make such contributions available to the community.

Additionally, out of curiosity, I noticed that the supplementary file contains two papers of unclear identity, labeled as being from ICLR. I would be grateful if you could clarify the intention behind including them.

---

> ### Author Rebuttal · Authors · 2026-03-31
>
> We thank the reviewer for their positive recommendation and acknowledging our work as “a well-written and commendable piece of research”. We address the remaining concerns below.
>
> > Are there any plans to release the training data or models?
>
> A: We appreciate the reviewer’s interest and fully agree that publicly available RAG-related models and datasets for the speech domain remain scarce. **We are actively working toward releasing both the AudioCRAG benchmark and the post-training data pipeline.** Specifically, we plan to release: (1) the AudioCRAG benchmark (both Synthetic and Human variants) and (2) the post-training scripts and data generation pipeline. We expect to make these available upon acceptance and will include a concrete release timeline in the camera-ready version.
>
> > The supplementary file contains two papers of unclear identity. Could you clarify?
>
> A: We apologize for the confusion. **The two papers included in the supplementary material are our own concurrent works on open-ended audio understanding and generation** (See S. A.16 for details). These works focus on expanding end-to-end audio capabilities but do not address tool-integration challenges, making them complementary to Stream RAG. We will add a README to the supplementary material in the revised version to clarify their purpose and relationship to the current submission.

---

> > ### Author Rebuttal · Reviewer_HhuX · 2026-04-03
> >
> > I’ll keep this score.

---

> > > ### Author Response · Authors · 2026-04-04
> > >
> > > We sincerely thank Reviewer HhuX for the positive recommendation. Thank you for your time and constructive support.

---

### Decision · Program_Chairs · 2026-04-30

**Decision:**

Accept (regular)

**Comment:**

By performing tool queries in parallel with user input, Stream RAG reduces latency in speech-based dialogue systems.
The architecture relies on three core elements: a triggering policy (utilizing Model-Triggering or Fixed-Intervals), a management framework that allows incoming queries to preempt active ones, and a matching system designed to identify the first valid result corresponding to the user's completed query.
The post-training pipeline teaches existing E2E speech-to-speech models when to invoke tools and how to generate spoken summaries from retrieved text, without full retraining.
Empirical results from the SLUE-SQA and the newly developed AudioCRAG benchmarks demonstrate that this approach improves both accuracy and system responsiveness.

As the first RAG implementation for full-duplex speech models, Stream RAG offers a robust solution to critical performance bottlenecks, providing improvements in latency while maintaining high precision.

Reviewer-Requested Modifications:
The human evaluation data shared during the rebuttal (100 samples reviewed by two annotators) should be fully incorporated into the main text.
The primary analysis needs to include a post-trained sequential RAG baseline, along with an in-depth assessment of its accuracy and efficiency.